# OVERTHINKING THE TRUTH: UNDERSTANDING HOW LANGUAGE MODELS PROCESS FALSE DEMONSTRATIONS

## ABSTRACT

Through few-shot learning or chain-of-thought prompting, modern language models can detect and imitate complex patterns in their prompt. This behavior allows language models to complete challenging tasks without fine-tuning, but can be at odds with completion quality: if the context is inaccurate or harmful, then the model may reproduce these defects in its completions. In this work, we show that this harmful context-following appears late in a model's computation–in particular, given an inaccurate context, models perform *better* after zeroing out later layers. More concretely, at early layers models have similar performance given either accurate and inaccurate few-shot prompts, but a gap appears at later layers (e.g. layers 13-14 for GPT-J). This gap appears at a consistent depth across datasets, and coincides with the appearance of "induction heads" that attend to previous answers in the prompt. We restore the performance for inaccurate contexts by ablating a small subset of these heads, reducing the gap by 23.2% on average across 14 datasets. Our results suggest that studying early stages of computation could be a promising strategy to prevent misleading outputs, and that understanding and editing internal mechanisms can help correct unwanted model behavior.

## 1 INTRODUCTION

A key behavior of modern language models is context-following: neural networks like GPT-3 are able to infer and imitate the patterns in their prompt. At its best, this allows language models to perform well on benchmarks without the need for fine-tuning (Brown et al., 2020; Rae et al., 2021; Hoffmann et al., 2022; Chowdhery et al., 2022; Srivastava et al., 2022). This has led researchers to study how properties of the context affect few-shot performance (Min et al., 2022b; Kim et al., 2022; Xie et al., 2021; Zhao et al., 2021), and what internal mechanisms underlie context-following (Olsson et al., 2022).

However, context-following can also lead to incorrect, toxic or unsafe model outputs (Rong, 2021). For example, if an inexperienced programmer prompts Codex (Chen et al., 2021) with poorly written or vulnerable code, the model is likely to produce poorly written or vulnerable code completions. Similarly, in this work we study few-shot learning for classification tasks: prompting the model with inaccurate demonstrations reduces model accuracy (Figure 1, left), because the model learns to reproduce the false demonstrations. We thus ask: Can we attribute this "false context-following" behavior to specific model components, and can we mitigate it by intervening on these components?

We show that, perhaps surprisingly, false context-following in text classification is primarily a property of late stages of computation. In particular, stopping the model early–by zeroing out the later layers (Nostalgebraist, 2020)–actually *improves* performance (Figure 1, center). Moreover, true and false contexts yield similar accuracy until some "critical layer" at which they sharply diverge. This demonstrates that even with false demonstrations, the model often "knows" the correct answer (it can be easily decoded from the latent states) but later replaces it with an incorrect answer that is more likely given the context.

To identify the underlying mechanism for false context-following, we turn to Olsson et al. (2022), who identify "induction heads" that attend to and reproduce previous patterns in the input. Motivated

by this, we searched for heads that consistently attend to previous examples that have the same (true) answer as the current prompt. We found many such heads, primarily concentrated in later layers of the model (after the critical layer). By removing 10 of these heads, we are able to reduce the accuracy gap between accurate and inaccurate prompts by an average of 23.2% over 14 datasets, with negligible effects on the performance given true prefixes (Figure 1, right).

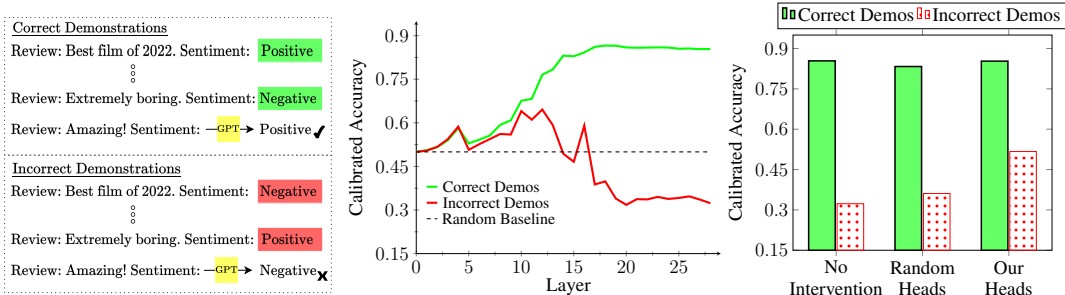

Figure 1: **Left:** Given a prompt of inaccurate demonstrations, language models are more likely to output incorrect labels. **Center:** When demonstrations are incorrect, zeroing out the later layers increases the classification accuracy, here on SST-2. **Right:** We identify 10 attention heads and remove them from the model: this reduces the effect of incorrect demonstrations by 36.7% on SST-2, averaged over 15 prompt formats, without decreasing the accuracy given correct demonstrations.

Our findings show how analyzing and editing model internals can help practitioners understand and mitigate model failures. Indeed, one intuition for why early-exiting succeeds is that the attention heads we identified cannot in general occur at the earliest layers. This is because these heads must recognize which inputs belong to the same class, which likely requires multiple layers of processing. Thus, early exiting might be a generally promising strategy to detect dishonest behavior in models.

## 2  PRELIMINARIES: FEW-SHOT LEARNING WITH FALSE DEMONSTRATIONS

We begin by introducing the setting we study: few-shot learning for classification, given demonstrations with correct or incorrect labels. Incorrect demonstrations consistently reduce classification performance, which is the phenomenon that we aim to study and mitigate in this work.

**Few-shot learning.** We consider autoregressive transformer language models, which produce a conditional probability distribution $p(t_{n+1} \mid t_1, ..., t_n)$ over the next token $t_{n+1}$ given previous tokens. We focus on the few-shot learning setting (Brown et al., 2020) for classification tasks: we sample $k$ demonstrations (input-label pairs) from the task dataset, denoted $(x_1, y_1), ..., (x_k, y_k)$. To query the model on a new input $x$, we use the predictive distribution $p(y \mid x_1, y_1, ..., x_k, y_k, x)$.

**Datasets and models.** We consider fourteen text classification datasets: SST-2 (Socher et al., 2013), Poem Sentiment (Sheng & Uthus, 2020), Financial Phrasebank (Malo et al., 2014), Ethos (Mollas et al., 2020), TweetEval-Hate (Barbieri et al., 2020), TweetEval-Atheism (Barbieri et al., 2020), TweetEval-Feminist (Barbieri et al., 2020), Medical Questions Pairs (McCreery et al., 2020), MRPC (Wang et al., 2019), SICK (Marelli et al., 2014), RTE (Wang et al., 2019), AGNews (Zhang et al., 2015), TREC (Voorhees & Tice, 2000), and DBpedia (Zhang et al., 2015). We used the same prompt formats as in Min et al. (2022b) and Zhao et al. (2021) (Table 2, 3). For SST-2 we use the first of the 15 prompt formats in Zhao et al. (Table 5). We evaluated 3 autoregressive language models: GPT-J (Wang & Komatsuzaki, 2021), GPT2-XL (Radford et al., 2019), and GPT-NeoX-20B (Black et al., 2022).

**Evaluation metrics.** Given our focus on classification tasks, we are interested in how often the model assigns higher probability to the true label than to all other labels. However, model predictions can be very unstable with respect to small prompt perturbations (Gao et al., 2021). To mitigate this variability, we measure the *calibrated* classification accuracy (Zhao et al., 2021). Concretely, for a 2-class classification task, we measure how often the correct label has a higher probability than its median probability over the dataset. Assuming the dataset is balanced (which is true for us), this step has been shown to improve performance and reduce variability across prompts. Calibration for multi-class tasks follows a similar procedure, detailed in appendix A.1.

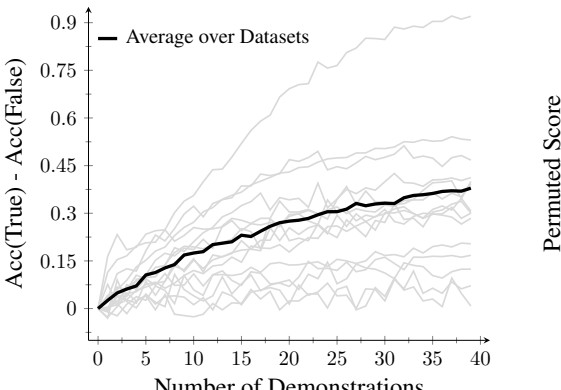 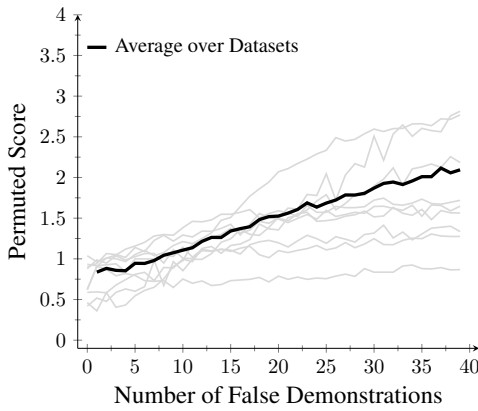

Figure 2: **Left:** The difference in accuracy between accurate and inaccurate prompts increases with the number of demonstrations. **Right:** As the number of false demonstrations increases, the model chooses the permuted label $\sigma(\text{class}(x))$ more often than the other labels, rather than making random errors.

## 2.1 COMPARING TRUE AND FALSE DEMONSTRATIONS

We first confirm that the models we study exhibit false context-following behavior. To do so, we compare the performance of models when the demonstration labels are all correct, i.e. $y_i = \text{class}(x_i)$, and when they are all incorrect, i.e. $y_i = \sigma(\text{class}(x_i))$, for a cyclic permutation $\sigma$ over the set of labels (Figure 1, left). In particular, inputs from the same class are always assigned the same (possibly false) label within each prompt.

For each model and dataset, we sample 1000 sequences each containing $k$ demonstrations and evaluate the model's calibrated accuracy. We sample different demonstrations $(x_i, y_i)$ and label permutations $\sigma$ for every sequence, and vary $k$ from 0 to 40 (from 0 to 20 for GPT2-XL, due to its smaller context size).

Figure 2 (left) shows the difference between GPT-J's calibrated accuracy given accurate and inaccurate prompts as the number of demonstrations increases. As expected, false demonstrations lead to worse performance, and the gap tends to increase with $k$ for most datasets. These results are in agreement with Min et al. (2022b), who found that incorrect demonstrations decreased GPT-J's performance on classification tasks (see Figure 4 in Min et al.).

Models could lose accuracy by copying the incorrect label, or by becoming confused and choosing random labels. To confirm it is the former, we also measure which labels the model chooses for multi-class tasks. Specifically, we measure the *permuted score*: how often the model chooses the permuted label $\sigma(\text{class}(x))$ over the other labels. For each dataset, a random classifier would have a permuted score of $\frac{1}{\#\text{labels}}$. To make the results comparable across datasets, we divide the permuted scores by this random baseline. Figure 2 (right) shows these reweighted permuted scores for GPT-J on the 9 multi-class datasets in our collection, as well as their average over the datasets. The permuted score increases steadily with the number of demonstrations and reaches twice its baseline value after 40 demonstrations.

## 3 THE LOGIT LENS: ZEROING OUT LATER LAYERS IMPROVES ACCURACY

In this section, we decode model predictions directly from intermediate layers. This allows us to evaluate the model's performance midway through processing the inputs. On false prefixes, we find that the model performs *better* midway through processing, and investigate this phenomenon in detail.

**Intermediate layer predictions: the logit lens.** Given an autoregressive transformer language model, we will decode a probability distribution over the next token from each intermediate layer,

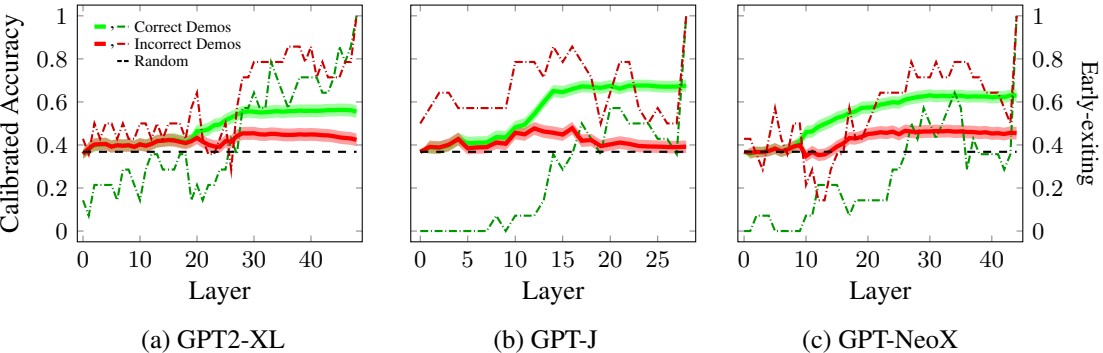

(a) GPT2-XL        (b) GPT-J        (c) GPT-NeoX

Figure 3: Average performance across 14 tasks for GPT2-XL, GPT-J, and GPT-NeoX. **y-axis (left):** Calibrated accuracy given correct and incorrect demonstrations, denoted by full lines. **y-axis (right):** Percentage of tasks where zeroing out all succeeding transformer blocks is superior than full model evaluation, denoted by dashed lines. Early-exiting is effective given false demonstrations, and perhaps more surprisingly, also effective given correct demonstrations.

using the "logit lens" method (Nostalgebraist, 2020). Intuitively, these intermediate distributions represent model predictions after $\ell \in \{1, ..., L\}$ layers of processing.

In more detail, let $h_\ell^{(i)} \in \mathbb{R}^d$ denote the hidden state of token $t_i$ at layer $\ell$, i.e. the sum of everything up to layer $\ell$ in the residual stream. For a sequence of tokens $t_1, ..., t_n \in V$, the logits of the predictive distribution $p(t_{n+1} \mid t_1, ..., t_n)$ are given by

$$[\text{logit}_1, ..., \text{logit}_{|V|}] = W_U \cdot \text{LayerNorm}_L(h_L^{(n)}),$$

where $\text{LayerNorm}_L$ is the the pre-unembedding layer normalization, and $W_U \in \mathbb{R}^{|V| \times d}$ is the unembedding matrix. The logit lens applies the same unembedding operation to the earlier hidden states $h_\ell^{(i)}$, yielding an intermediate layer distribution $p_\ell(t_{n+1} \mid t_1, ..., t_n)$:

$$[\text{logit}_1^\ell, ..., \text{logit}_{|V|}^\ell] = W_U \cdot \text{LayerNorm}_L(h_\ell^{(n)}).$$

This provides a measurement of what predictions the model represents at layer $\ell$, without the need to train a new decoding matrix. It can therefore be interpreted as a form of early exiting (Panda et al., 2015; Teerapittayanon et al., 2017; Figurnov et al., 2017).

**Early exiting improves classification performance.** We measure the calibrated accuracies of the intermediate layer distributions $p_\ell$ for the three models and fourteen datasets from Section 2, using context lengths of 40 demonstrations (20 demonstrations for GPT2-XL). We also measure the layerwise accuracies for two toy datasets: "SST-2-A/B", a modification of SST-2 (Socher et al., 2013), and "Unnatural", that extends a task in Rong (2021; section 4). In "SST-2-A/B", we replace the labels (e.g. 'Positive' and 'Negative') with letters 'A' and 'B'. In "Unnatural", demonstrations are of the form "[object]: [label]" and the labels are "plant/vegetable", "sport", and "animal".

Figure 4 displays results for GPT-J, with corresponding plots for GPT2-XL and GPT-NeoX in Figures 9 and 10 in the Appendix. For GPT-J with correct demonstrations, accuracy tends to increase with layer depth, and starts to stagnate or grow more slowly around layer 15. The accuracy for incorrect demonstrations follows a similar trend at the early layers, but then diverges and decreases at the later layers.

For incorrect demonstrations, decoding from earlier layers performs *better* than decoding from the final layer. For GPT-J, using $p_{16}$ (the first 16 layers) achieves a better accuracy than the full model for 12 out of the 14 datasets, by an average of 8.6 percentage points. This gain on false prefixes comes with a comparatively small cost for true prefixes: 1.6 percentage points (see Table **??**). Similarly, for GPT2-XL and GPT-NeoX, the intermediate predictions $p_{30}$ and $p_{27}$ outperform using the full model for 11 out of 14 datasets, however the magnitude of the effect is smaller: an average of 3 percentage points for GPT2-XL and 0.7 percentage points for GPT-NeoX.

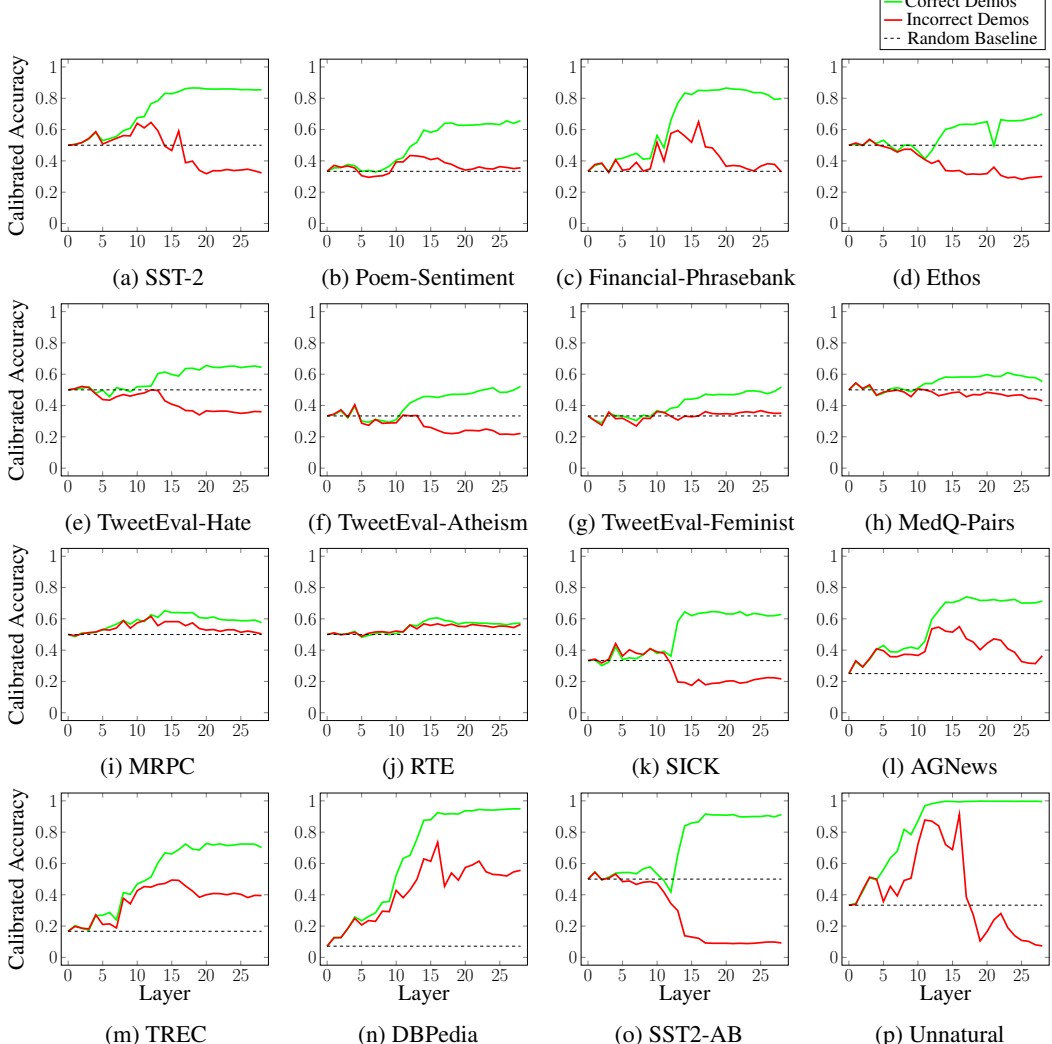

Figure 4: GPT-J early-exit classification accuracies across 16 tasks, given correct and incorrect demonstrations. Plots are grouped by task type: sentiment analysis (a-c), hate speech detection (d-g), paraphrase detection (h-i), natural language inference (j-k), topic classification (l-n), and toy tasks (o-p). Given incorrect demonstrations, zeroing out all transformer blocks after layer 16 outperforms running the entire model, across 14 out of the 16 tasks.

For the toy "Unnatural" dataset, these effects are particularly pronounced (see Figure 4p). At layer 16, the accuracy of GPT-J for incorrect demonstrations reaches 0.91, which is 92% of the final layer accuracy given an accurate prompt. In contrast, at the final layer, GPT-J's accuracy given false demonstrations reaches its lowest value, 0.07.

**True and false prefixes sharply diverge at "critical layers".** For each model, the accuracies for correct and incorrect demonstrations diverge at the same layers across all datasets. For example, for GPT-J, the accuracy gap between accurate and inaccurate prompts first exceeds 50% (and 45%, and 55%) of its final layer value between layers 13 and 14 for 12 out of the 14 datasets. We obtain similar results for GPT-NeoX with layers 10 to 12 and for GPT2-XL with layers 21 to 23. In summary, zeroing out later layers leads to better classification accuracies given incorrect demonstrations, and the accuracy gap between correct and incorrect demonstrations emerges at a consistent set of layers across datasets.

Figure 5: Examples of attention patterns on incorrect demonstrations from the "unnatural" dataset, for heads that are label-attending but not class-sensitive (Left), heads that are class-sensitive but not label-attending (Center), and heads that are both label-attending and class-sensitive (Right).

## 4 ZOOMING INTO ATTENTION HEADS

We found that for all datasets, the gap between true and false demonstrations appears in a small set of transformer blocks. We would like to know whether some specific attention heads are responsible for this behavior.

(Olsson et al., 2022) introduce *induction heads*: attention heads that attend to previous occurences of the present token, and increase the probability of the outputs that follow them. Inspired by this work, we investigate the hypothesis that a small number of "induction heads" play a key role in false context-following, by attending to the labels in previous similar demonstrations and making the model more likely to output them.

For example, in Figure 5, we know that the model assigns a high probability to the mistaken label "sport". According to the hypothesis, this is because of heads that attend to the previous occurences of "sport" in this context, and increase the probability of that token. The previous occurences of "sport" share two properties: (1) they are *labels* in the previous demonstrations, and (2) they follow inputs *with the same class* as "beet": "tomato" and "garlic".

Therefore, we look for heads that satisfy two conditions when they attend to inaccurate prompts. First, they should be *label-attending*, i.e. concentrate their attention on labels in the previous demonstrations. Second, they should be *class-sensitive*, meaning they should attend specifically to those labels that follow inputs in the same class as the latest input. We call heads that are both label-attending and class-sensitive given incorrect demonstrations *false prefix-matching heads*.

We define a score to identify false prefix-matching heads. For a sequence of demonstrations $(x_i, y_i)$ and a final input $x$, the **prefix-matching score** ($\text{PMS}^h$) of a head $h$ is:

$$\text{PMS}^h = \sum_{i=1}^n \text{Att}^h(x, y_i) \cdot \mathbf{1}_{\text{class}(x)=\text{class}(x_i)} - \frac{1}{\#\text{labels} - 1} \sum_{i=1}^n \text{Att}^h(x, y_i) \cdot \mathbf{1}_{\text{class}(x)\neq\text{class}(x_i)}.$$

Prefix-matching heads should have a high PMS scores. We therefore plot the distribution of prefix-matching scores across layers for these heads. Figure 6 shows these results for the "Unnatural" dataset. For each model, the scores remain low at the early layers, but then increase around the "critical layers" that we identified in the previous section. This lends correlational support to our hypothesis that false prefix-matching heads cause false-context following behavior.

**Ablating false prefix-matching heads.** However, we are interested in causal evidence. Therefore, we check whether removing false prefix-matching heads reduces false context-following. We select the 10 heads from GPT-J with the highest prefix-matching scores given incorrect demonstrations on the "Unnatural" dataset. We ablate these heads by setting their keys, queries and values to zero. We then evaluate the resulting lesioned model on all 14 datasets, and compare its layerwise performance to the original model's. As a control baseline, we also perform the same analysis for 10 heads selected at random.

The ablations considerably increase the accuracy given false demonstrations: they reduce the gap in accuracy between accurate and inaccurate prompts by an average of 23.2% for $k = 40$ and 39.3%

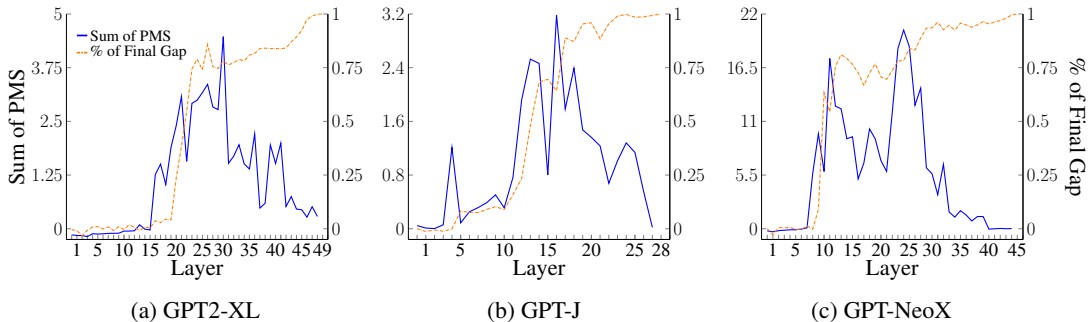

(a) GPT2-XL       (b) GPT-J       (c) GPT-NeoX

Figure 6: Sum of prefix-matching scores given true and false demonstrations, for GPT2-XL (a), GPT-J (b), and GPT-NeoX (c) on the toy Unnatural dataset. The prefix-matching scores increase around the layers where the accuracy gap (averaged over tasks) between true and false demonstrations emerges.

for $k = 10$ (see Table 1). In contrast, ablating random heads reduces the gap by 3.6% for $k = 40$ and 13.4% for $k = 10$. While they greatly improve the accuracy given a false prefix, our ablations have a comparatively small effect on the accuracy given correct demonstrations: ablating the false prefix-matching heads decreases the accuracy given true demonstrations by 2.2% for $k = 40$ and by 1.3% for $k = 10$. These results show that the false prefix-matching heads cause a large fraction of the false context-following behavior.

**Analysing the outputs of false prefix-matching heads.** We identified false prefix-matching heads based only on their attention patterns. However, our postulated mechanism also depends on the heads' outputs: they must increase the probability of the labels they attend to. We therefore study the outputs of these heads to understand how they affect the residual stream.

We apply the logit lens to each head individually, by applying layer normalization followed by the unembedding matrix to its outputs. This tells us how much the head increases or decreases the intermediate logits of each token. For every head, we measure the difference between the logit increases of the permuted and correct labels on the Unnatural dataset (following the methodology in (Wang et al., 2022)). Our 10 false prefix-matching heads have an average score of 1.2, which shows that they increase the logits of the permuted label more than those of the correct label. In contrast, when sampling 100 sets of 10 random heads, we find an average score of 0.25, with a standard deviation of 0.4. Therefore, false prefix-matching heads directly increase the probability of the permuted labels relative to the correct labels more often than random heads.

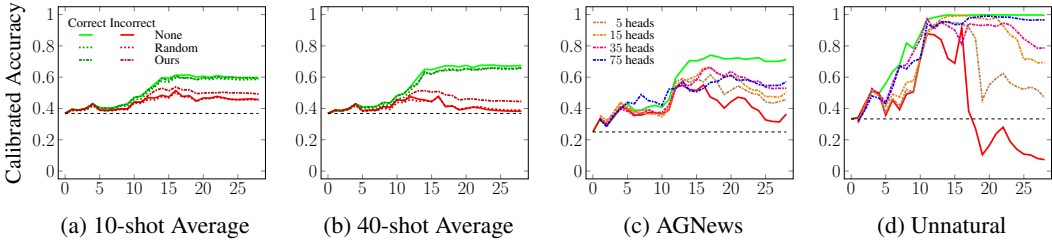

(a) 10-shot Average    (b) 40-shot Average    (c) AGNews    (d) Unnatural

Figure 7: Ablating false prefix-matching heads increases accuracy across multiple layers. **(a), (b)**: Average accuracy at each layer before and after ablating false-prefix matching or random heads, given correct and incorrect demonstrations. **(c), (d)**: Accuracy at each layer for incorrect demonstrations on AGNews and Unnatural, after ablating the $k$ most class-sensitive heads, for $k \in \{5, 15, 35, 75\}$.

## 5   DISCUSSION AND RELATED WORK

In this paper, we showed how stopping language models early by zeroing out their later layers improves classification performance given inaccurate contexts, without requiring any additional train-

Table 1: Ablating false prefix-matching heads recovers a large fraction of the accuracy gap between true and false prefixes, without hurting performance given true prefixes. We show the percentage reduction of the accuracy gap and percentage change in true prefix performance when ablating the 10 false prefix-matching heads chosen using the Unnatural dataset ("top") or 10 random heads ("random"). We bold gap reductions when they are greater for our heads than for the random heads.

| Dataset | Heads | 10-shot | | 40-shot | |
|---|---|---|---|---|---|
| | | True Prefix %$\Delta$($\uparrow$) | Gap Reduction %($\uparrow$) | True Prefix %$\Delta$($\uparrow$) | Gap Reduction %($\uparrow$) |
| *Sentiment Analysis* | | | | | |
| SST-2 | top | $0.49_{0.04}$ | $\mathbf{33.66}_{0.22}$ | $-0.09_{0.05}$ | $\mathbf{36.69}_{0.44}$ |
| | random | $-2.34_{0.08}$ | $7.15_{0.05}$ | $-2.51_{0.11}$ | $11.18_{0.04}$ |
| Poem-Sentiment | top | $2.51_{0.07}$ | $\mathbf{27.05}_{0.13}$ | $0.46_{0.01}$ | $\mathbf{33.59}_{0.23}$ |
| | random | $-5.41_{0.16}$ | $14.21_{0.11}$ | $-5.28_{0.16}$ | $13.19_{0.08}$ |
| Financial-Phrasebank | top | $-2.78_{0.08}$ | $\mathbf{30.04}_{0.11}$ | $-3.01_{0.11}$ | $\mathbf{24.96}_{0.16}$ |
| | random | $-2.83_{0.09}$ | $9.08_{0.04}$ | $-4.22_{0.15}$ | $9.34_{0.09}$ |
| *Hate Speech Detection* | | | | | |
| Ethos | top | $-7.36_{0.21}$ | $\mathbf{43.16}_{0.01}$ | $-4.86_{0.15}$ | $\mathbf{21.50}_{0.05}$ |
| | random | $-1.34_{0.04}$ | $8.42_{5.13}$ | $-2.86_{0.09}$ | $10.5_{0.01}$ |
| TweetEval-Hate | top | $-3.43_{0.10}$ | $10.20_{0.07}$ | $-3.42_{0.10}$ | $\mathbf{15.49}_{0.01}$ |
| | random | $-4.05_{0.12}$ | $11.22_{0.08}$ | $-0.31_{0.01}$ | $9.15_{0.06}$ |
| TweetEval-Atheism | top | $-2.59_{0.07}$ | $19.77_{0.01}$ | $-5.30_{0.14}$ | $\mathbf{14.43}_{0.02}$ |
| | random | $-3.07_{0.09}$ | $25.21_{0.01}$ | $-2.49_{0.06}$ | $7.66_{0.01}$ |
| TweetEval-Feminist | top | $2.35_{0.07}$ | $\mathbf{4.15}_{0.07}$ | $1.09_{0.03}$ | $6.16_{0.07}$ |
| | random | $-0.30_{0.01}$ | $2.86_{0.02}$ | $-3.60_{0.10}$ | $16.10_{0.03}$ |
| *Paraphrase Detection* | | | | | |
| MedQ-Pairs | top | $-3.70_{0.11}$ | $\mathbf{56.67}_{0.01}$ | $-10.83_{0.30}$ | $\mathbf{56.45}_{0.07}$ |
| | random | $-2.22_{0.06}$ | $46.67_{0.01}$ | $-4.69_{0.14}$ | $38.71_{0.01}$ |
| MRPC | top | $-0.38_{0.01}$ | $\mathbf{99.50}_{0.01}$ | $-0.35_{0.01}$ | $\mathbf{41.67}_{0.05}$ |
| | random | $-1.14_{0.03}$ | $22.22_{0.01}$ | $-1.74_{0.05}$ | $25.00_{0.01}$ |
| *Natural Language Inference* | | | | | |
| SICK | top | $-12.82_{0.35}$ | $\mathbf{38.40}_{0.11}$ | $-10.34_{0.30}$ | $\mathbf{29.51}_{0.01}$ |
| | random | $-2.77_{0.07}$ | $11.13_{0.01}$ | $-7.85_{0.22}$ | $16.41_{0.08}$ |
| RTE | top | $8.77_{0.27}$ | $\mathbf{81.81}_{0.05}$ | $4.20_{0.13}$ | $\mathbf{-25.00}_{0.19}$ |
| | random | $1.75_{0.05}$ | $45.45_{0.02}$ | $0.35_{0.01}$ | $-99.99_{0.01}$ |
| *Topic Classification* | | | | | |
| AGNews | top | $1.62_{0.05}$ | $\mathbf{21.37}_{0.13}$ | $0.64_{0.02}$ | $\mathbf{32.20}_{0.27}$ |
| | random | $-0.59_{0.02}$ | $-18.80_{0.12}$ | $-2.57_{0.09}$ | $5.26_{0.08}$ |
| TREC | top | $0.16_{0.01}$ | $\mathbf{49.03}_{0.09}$ | $1.00_{0.03}$ | $\mathbf{19.22}_{0.19}$ |
| | random | $-4.30_{0.13}$ | $7.69_{0.13}$ | $-0.28_{0.01}$ | $-5.21_{0.07}$ |
| DBPedia | top | $-1.21_{0.06}$ | $\mathbf{28.70}_{0.17}$ | $0.63_{0.05}$ | $\mathbf{18.32}_{0.40}$ |
| | random | $-1.32_{0.07}$ | $-5.21_{0.25}$ | $-0.84_{0.06}$ | $-6.62_{0.26}$ |
| Average | top | $-1.31_{0.49}$ | $\mathbf{39.26}_{0.76}$ | $-2.16_{0.79}$ | $\mathbf{23.23}_{1.95}$ |
| | random | $-2.14_{0.92}$ | $13.38_{0.79}$ | $-2.78_{1.24}$ | $3.62_{0.59}$ |

ing. We also identified attention heads that contribute to the effect of misleading prompts, and showed that ablating these heads mitigates this effect.

**Related work.** Our work is closely related to Min et al. (2022b) and Kim et al. (2022), who examine the role of false demonstration on model accuracy. Min et al. (2022b, figure, 4) find that for classification by a pre-trained model (GPT-J), the ground truth of demonstrations has a large effect on the accuracy. However, they do not find such an effect with a meta-tuned model (Min

et al., 2022a). Therefore, meta-tuning could serve as "negative control" to test our hypothesis that false prefix-matching heads cause false context-following: an interesting future direction would be to check whether meta-tuning reduces the number of false prefix-matching heads.

The literature on early-exiting and overthinking (Kaya et al., 2018; Panda et al., 2015; Teerapittayanon et al., 2017; Figurnov et al., 2017; Hou et al., 2020; Liu et al., 2020; Xin et al., 2020; Zhou et al., 2020; Zhu, 2021; Schuster et al., 2022) also highlights how decoding from intermediate layers can save compute and sometimes produce better results. One major difference is that most of these methods rely either on modifying the training process to allow for early-exit, or on training additional probes to decode intermediate states. In contrast, the logit lens does not require any extra training to decode answers from internal representations.

**How does the logit lens compare to probing?** Our work, especially Section 3, relies heavily on the "logit lens" (Nostalgebraist, 2020). We find it useful to think of this method in comparison to probing.

If a layer has a high probing accuracy, this means that the correct answer can be decoded from the hidden states. However, this is often a low bar to clear, especially when the classification task is easy and the hidden states are high-dimensional (Hewitt & Liang, 2019). In contrast, if a layer has a high logit lens accuracy, this shows that it encodes correct answers along a direction in the residual stream that the model subsequently decodes from, which is much more informative. On the other hand, a low logit lens performance at a layer does not imply that the correct answers cannot be decoded from that layer.

One intermediate between probing and zeroing out later layers is the "tuned lens" (Ostrovsky et al., 2022): instead of training probes on each classification task or directly using the final layer's decoding matrix, for each layer the authors train a new square adapter matrix between the residual stream and the unembedding matrix on a language modelling dataset such as the Pile (Gao et al., 2020). It would be interesting to run our experiments with this alternative decoding method.

**Future work.** While we find consistent results across 14 datasets, our experiments are restricted to a specific setting: text classification with a large number of incorrect few-shot examples. We also studied variations on our setting, with qualitatively similar results (Figure 14): when the demonstration labels are selected at random rather than according to a permutation, and when only half of the demonstration labels are incorrect. In the future, researchers could use the logit lens to study diverse real-world failures of large models, such as "prompt injection" (Branch et al., 2022) or vulnerable code completions (Pearce et al., 2022). In both of these cases, the model outputs inaccurate or harmful completions even though it is capable of producing correct ones given a better prompt.

In addition, our head ablations do not recover the entirety of the accuracy gap between accurate and inaccurate prompts. This could be because we did not identify some of the model components that cause false context-following. However, there is another possibility: if an attention head's outputs are on average far from zero, zeroing out that head takes the intermediate states off-distribution, which can decrease overall model performance. Thus, one promising future direction would be to replace head outputs by their value on different inputs, as in (Meng et al., 2022).

Relatedly, while we identified induction heads that increase the probability of the wrong answers, we still do not have a full mechanistic understanding of false context-following behavior. For example, we do not know which heads in the earlier layers compose to form these induction heads. Future work could build on our methodology to reverse-engineer circuits (Cammarata et al., 2020) in GPT models that implement false context-following.

In this work we showed how studying the early stages of computation can mitigate the effects of misleading prompts. We hope this will spur further work on auditing network internals to detect dishonest behavior in models.

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

# A APPENDIX

## A.1 CALIBRATION

For $k$-way tasks, we measure how often the correct label has a higher probabilitiy than the $\frac{k-1}{k}$-quantile of its probability over the dataset. In figure 12, we show the logit lens accuracies of GPT-J over the 16 datasets, and confirm that they are similar to the accuracies with calibration, albeit a bit noisier.

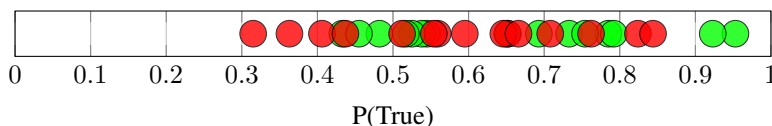

P(True)

Figure 8: The probability of the label "True" for 30 random test inputs in MRPC. The "True" class is marked with green dots and the "False" class is marked with red dots. As observed in Zhao et al. (2021), the model can be biased towards one of the labels.

## A.2 LOGIT LENS RESULTS FOR THE OTHER MODELS

We plot the Logit Lens results for GPT2-XL and GPT-NeoX in Figure 9 and Figure 10.

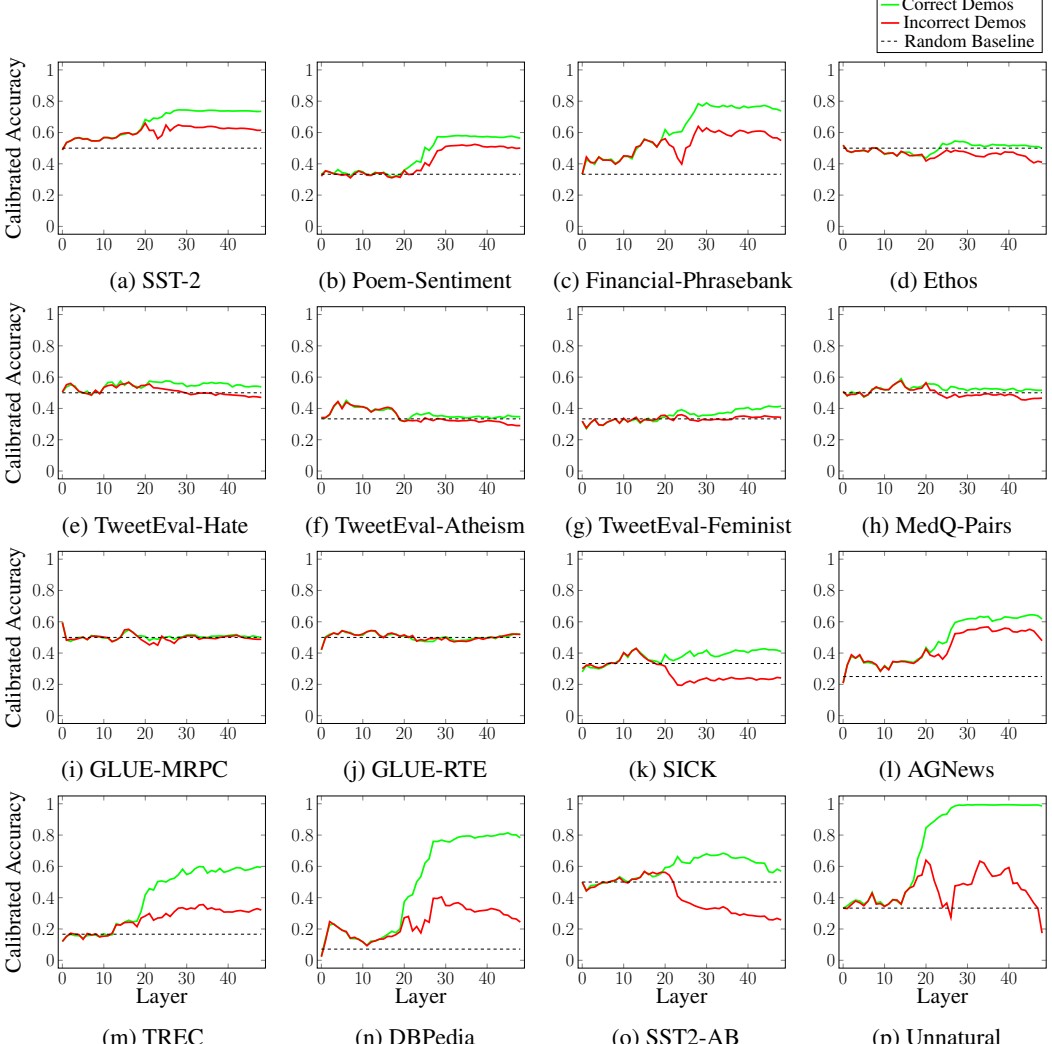

Figure 9: GPT2-XL early-exit classification accuracies across 16 datasets, given correct and incorrect demonstrations. Given incorrect demonstrations, zeroing out all transformer blocks after layer 30 outperforms running the entire model on 13 out of 16 datasets. In all datasets, running the entire model is never superior than the max performance over the preceding layers.

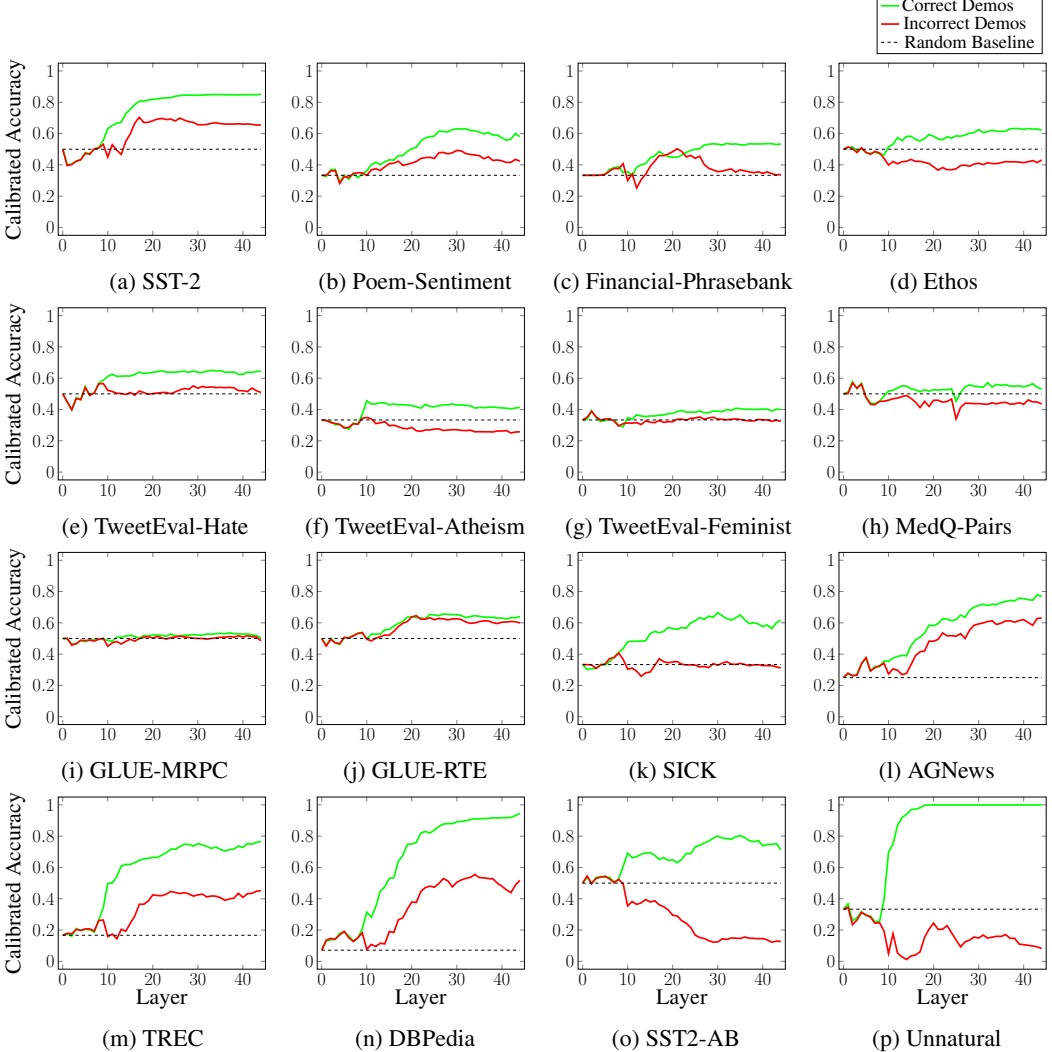

Figure 10: GPT-NeoX early-exit classification accuracies across 16 datasets, given correct and incorrect demonstrations. Given incorrect demonstrations, zeroing out all transformer blocks after layer 27 outperforms running the entire model on 13 out of 16 datasets. In all datasets, running the entire model is never superior than the max performance over the preceding layers.

## A.3 Logit lens results for each SST-2 prompt format

Figure 11: Calibrated Accuracy for all 15 prompt formats for SST-2 (from Zhao et al. (2021)). Given incorrect demonstrations, prompt formats 1, 2, 3, 4, 5, 7, 8, 9, 10, and 13 experience an increase in performance before experiencing a decline. Prompt formats 6, 12, 14, and 15, on the other hand, do not exhibit this effect. Prompt format 11 produces poor performance, given both correct and incorrect demonstrations. See Table 5 for prompt format details.

## A.4 LOGIT LENS RESULTS FOR GPT-J WITHOUT CALIBRATION

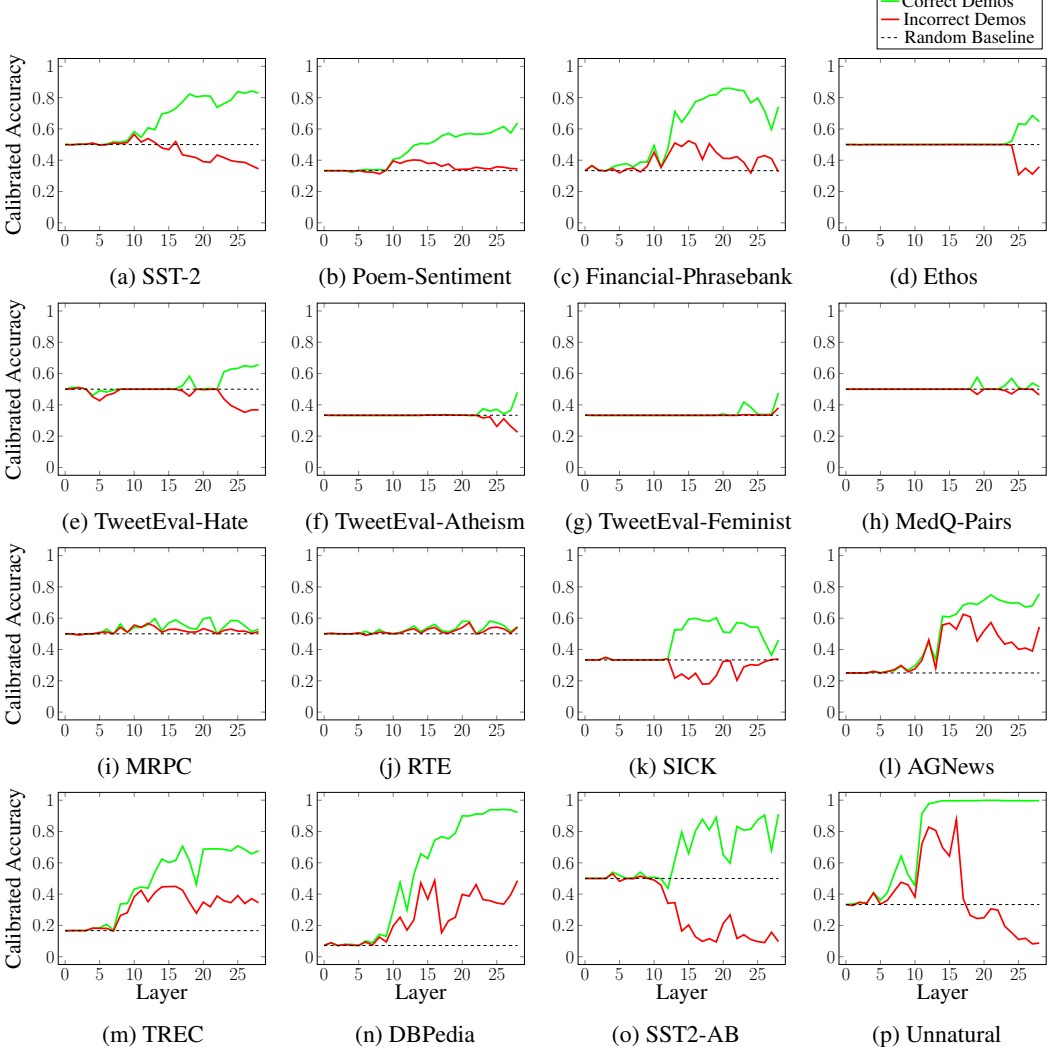

Figure 12: GPT-J early-exit *uncalibrated* classification accuracies across 16 tasks, given correct and incorrect demonstrations. The lack of calibration makes the results noisier especially at early layers, but early-exit still generally outperforms running the full model.

## A.5  LOGIT LENS RESULTS AS MEASURED BY OTHER METRICS

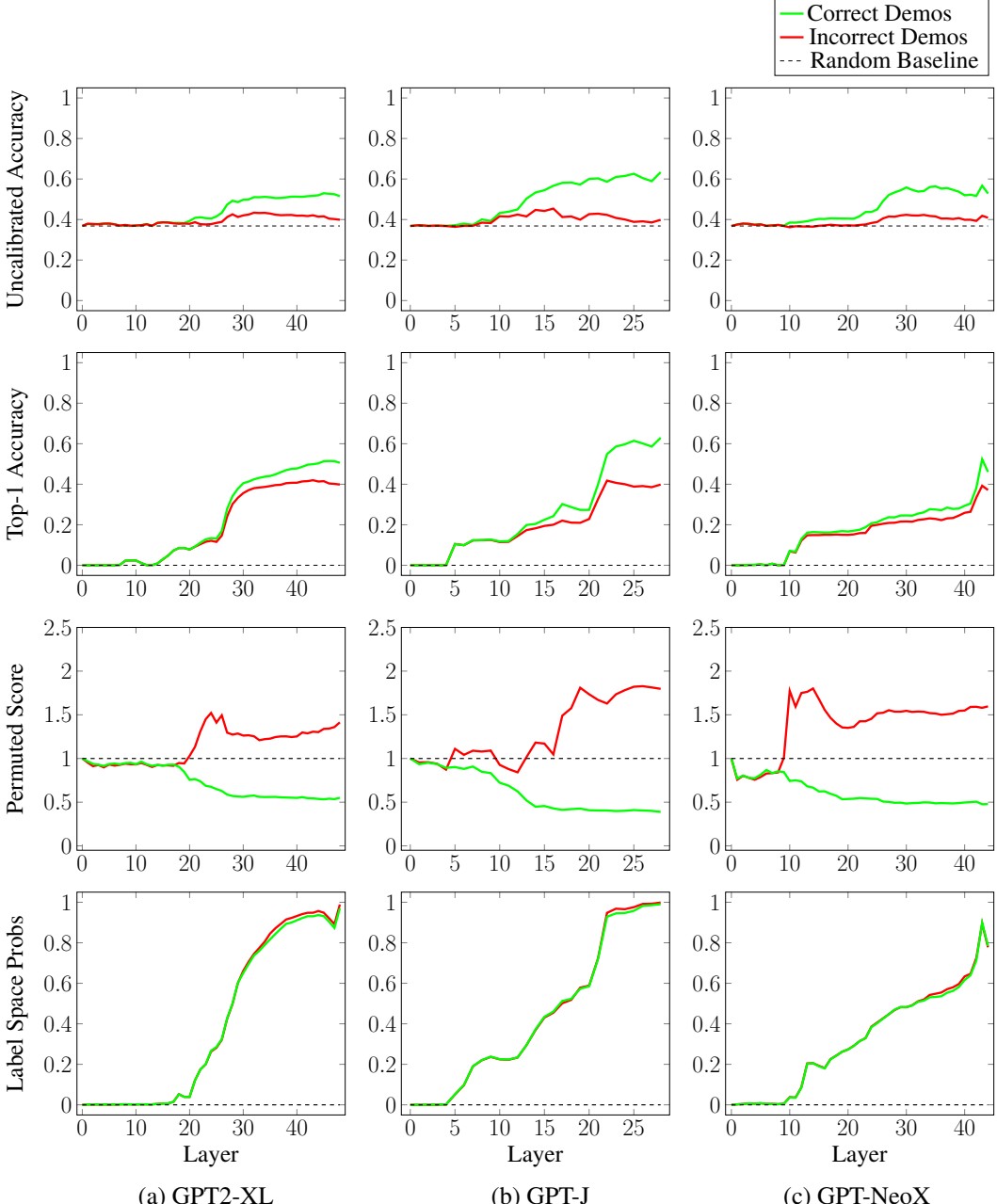

Figure 13: Uncalibrated Accuracy (**row 1**), Top-1 Accuracy (**row 2**), Permuted Score (**row 3**), and Label Space Probabilities (**row 4**) averaged over 14 tasks (9 multi-class tasks for the permuted score). As the label space is learned, we observe the emergence and ensuing increase in the gap in the other metrics.

## A.6 LOGIT LENS RESULTS FOR VARIANTS OF OUR SETUP

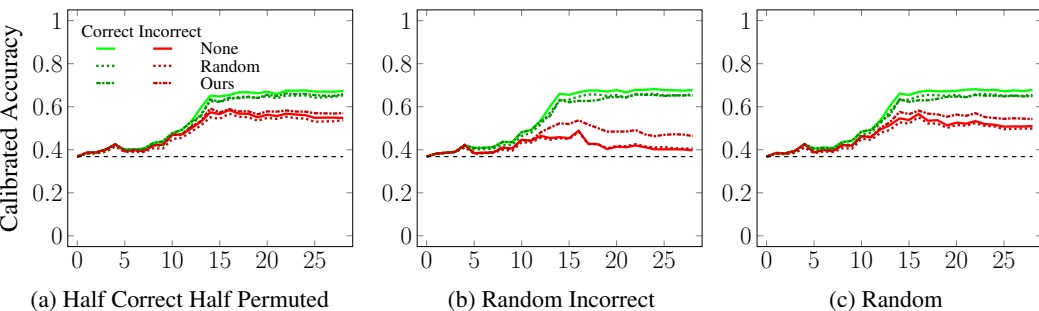

(a) Half Correct Half Permuted      (b) Random Incorrect      (c) Random

Figure 14: Average GPT-J layerwise accuracies for the original model, the null ablation, and our ablation in variants of our setup: when half of the 40 demonstrations are true and half have permuted labels (a), when each demonstration's label is chosen at random among the incorrect labels (b), and when each demonstration's label is chosen at random among all the labels (c). We find qualitatively similar results in these different settings.

A.7 PROMPT FORMATS USED FOR ALL DATASETS

Table 2: The prompts used for sentiment analysis and hate speech detection. The prompt used for SST-2 is taken from Zhao et al. (2021), and the prompts used for Poem-Sentiment, Financial-Phrasebank, Ethos, TweetEval-Hate, TweetEval-Atheism, and TweetEval-Feminist are taken from Min et al. (2022b). We show one training example per task for illustration.

| Task | Prompt | Labels |
|---|---|---|
| SST-2 | Review: Well-rounded tribute.
Answer: Positive.

Review: Saw how bad this movie was.
Answer: | Positive, Negative |
| Poem-Sentiment | it shines superior on a throne of gold:
The sentiment is: positive.

while the rude winds blow off each shadowy crown:
The sentiment is: | positive, negative, no impact |
| Financial-Phrasebank | Text: Operating profit for the nine-month period increased from EUR13 .6m, while net sales increased from EUR394 .7m, as compared to the corresponding period in 2005.
Sentiment: Positive.

Text: According to Gran, the company has no plans to move all production to Russia, although that is where the company is growing.
Sentiment: | Positive, Negative, Neutral |
| Ethos | Is the following hate speech? Answer yes or no.

Text: You look like Russian and speak like Indian. Both are disgusting go kill yourself.
Answer: yes.

Text: Civil rights movement for transgendered people
Answer: | yes, no |
| TweetEval-Hate | Tweet: Hundreds of Syrian refugees return home from Lebanon - ABC News.
Sentiment: favor.

Tweet: And now another flood of immigrants coming our way. #BuildThatWall
Sentiment: | favor, against |
| TweetEval-Atheism | Determine if the text supports atheism. Answer with yes, no, or neither.

Tweet: It's Ask an Atheist Day! Have a question? #askanatheist #SemST
Answer: yes.

Tweet: Oh Jesus, We write songs to praise you. #Songwriters #wewrite #Songs #Praiseyou #SemST
Answer: | yes, no, neither |
| TweetEval-Feminist | Determine if the text supports feminism. Answer with yes, no, or neither.

Tweet: FINALLY A WOMEN RUNNING FOR PRESIDENT #SemST
Answer: yes.

Tweet: Australia even has a fucking Minister for women for fucks sake! IsAwful #SemST
Answer: | yes, no, neither |

Table 3: The prompts used for paraphrase detection, natural language inference, and topic classification. The prompts for MedQ-Pairs, MRPC, SICK, and RTE are taken from Min et al. (2022b), and the prompt for AGNews, TREC, and DBPedia are taken from Zhao et al. (2021). We show one training example per task for illustration.

| Task | Prompt | Labels |
|------|--------|--------|
| MedQ-Pairs | Determine if the two questions are equivalent or not. | equivalent, not |
| | Question: After how many hour from drinking an antibiotic can I drink alcohol? Question: I have a party tonight and I took my last dose of Azithromycin this morning. Can I have a few drinks? Answer: equivalent. | |
| | Question: After how many hour from drinking an antibiotic can I drink alcohol? Question: I vomited this morning and I am not sure if it is the side effect of my antibiotic or the alcohol I took last night...? Answer: | |
| MRPC | The DVD-CCA then appealed to the state Supreme Court. The question is: The DVD CCA appealed that decision to the U.S. Supreme Court? True or False? The answer is: True. | True, False |
| | The Nasdaq composite index increased 10.73, or 0.7 percent, to 1,514.77. The question is: The Nasdaq Composite index, full of technology stocks, was lately up around 18 points? True or False? The answer is: | |
| SICK | The young boys are playing outdoors and the man is smiling nearby. The question is: The kids are playing outdoors near a man with a smile? True or False? The answer is: True. | True, False, Not sure |
| | Two people are kickboxing and spectators are not watching. The question is: Two people are kickboxing and spectators are watching? True or False? The answer is: | |
| RTE | The Armed Forces Press Committee (COPREFA) admitted that the government troops sustained 11 casualties in these clashes, adding that they inflicted three casualties on the rebels. The question is: Three rebels were killed by government troops? True or False? The answer is: True. | True, False |
| | Gastrointestinal bleeding can happen as an adverse effect of non-steroidal anti-inflammatory drugs such as aspirin or ibuprofen. The question is: Aspirin prevents gastrointestinal bleeding. True or False? The answer is: | |
| AGNews | Article: Bush, Republicans Outpoll Kerry, Democrats on TV (Reuters) Reuters - Although the election is not until. Answer: World. | World, Sports, Business, Science |
| | Article: Baseball Today (AP) AP - Chicago at Montreal (7:05 p.m. EDT). Greg Maddux (12-8) starts for the Cubs. Answer: | |
| TREC | Classify the questions based on whether their answer type is a Number, Location, Person, Description, Entity, or Abbreviation. | Description, Entity, Abbreviation, Person, Number, Location |
| | Question: What are liver enzymes? Answer Type: Description. | |
| | Question: What is considered the costliest disaster the insurance industry has ever faced? Answer Type: | |
| DBPedia | Classify the documents based on whether they are about a Company, School, Artist, Athlete, Politician, Transportation, Building, Nature, Village, Animal, Plant, Album, Film, or Book. | Company, School, Artist, Athlete, Politician, Transportation, Building, Nature, Village, Animal, Plant, Album, Film, Book |
| | Article: CIB Bank is the second-biggest commercial bank in Hungary after the 1 January 2008 merger with Inter-Európa Bank. This follows the 2007 merger of their respective Italian parent companies Banca Intesa and Sanpaolo IMI to form Intesa Sanpaolo. Answer: Company. | |
| | Article: Adarsh Vidya Kendra is a school in India. Answer: | |

Table 4: The prompts used for the toy tasks: Unnatural and SST-2-A/B. The prompt for Unnatural is taken from Rong (2021) and the prompt for SST-2-A/B is taken from the SST-2 prompt in Zhao et al. (2021). We show two training examples per task for illustration.

| Task | Prompt | Labels |
|---|---|---|
| SST-2-A/B | Review: Well-rounded tribute.
Answer: A.

Review: Saw how bad this movie was.
Answer: B.

Review: Skip this dreck.
Answer: | A, B |
| Unnatural | Consider the categories plant/vegetable, sport, and animal. Classify each object in its category.

onions: plant/vegetable.

hockey: sport.

horse: | animal, plant/vegetable, sport |

Table 5: The different prompt formats used for SST-2 from Zhao et al. (2021). We show one training example for illustration.

| Format ID | Prompt | Labels |
|---|---|---|
| 1 | Review: Well-rounded tribute.
Answer: Positive.

Review: Saw how bad this movie was.
Answer: | Positive, Negative |
| 2 | Review: Well-rounded tribute.
Answer: good.

Review: Saw how bad this movie was.
Answer: | good, bad |
| 3 | My review for last night's film: Well-rounded tribute. The critics agreed that this movie was good.

My review for last night's film: Saw how bad this movie was. The critics agreed that this movie was | good, bad |
| 4 | Here is what our critics think for this month's films.

One of our critics wrote "Well-rounded tribute." Her sentiment towards the film was positive.

One of our critics wrote "Saw how bad this movie was." Her sentiment towards the film was | positive, negative |
| 5 | Critical reception [ edit ]

In a contemporary review, Roger Ebert wrote "Well rounded tribute." Entertainment Weekly agreed, and the overall critical reception of the film was good.

In a contemporary review, Roger Ebert wrote "Saw how bad this movie was." Entertainment Weekly agreed, and the overall critical reception of the film was | good, bad |
| 6 | Review: Well rounded tribute.
Positive Review? Yes.

Review: Saw how bad this movie was.
Positive Review? | Yes, No |
| 7 | Review: Well rounded tribute.
Question: Is the sentiment of the above review Positive or Negative?
Answer: Positive.

Review: Saw how bad this movie was.
Question: Is the sentiment of the above review Positive or Negative?
Answer: | Positive, Negative |
| 8 | Review: Well rounded tribute.
Question: Did the author think that the movie was good or bad?
Answer: good.

Review: Saw how bad this movie was.
Question: Did the author think that the movie was good or bad?
Answer: | good, bad |
| 9 | Question: Did the author of the following tweet think that the movie was good or bad?
Tweet: Well rounded tribute.
Answer: good.

Question: Did the author of the following tweet think that the movie was good or bad?
Tweet: Saw how bad this movie was.
Answer: | good, bad |
| 10 | Well rounded tribute. My overall feeling was that the movie was good.

Saw how bad this movie was. My overall feeling was that the movie was | good, bad |
| 11 | Well rounded tribute. I liked the movie.

Saw how bad this movie was. I | liked, hated |
| 12 | Well rounded tribute. My friend asked me if I would give the movie 0 or 5 stars, I said 5.

Saw how bad this movie was. My friend asked me if I would give the movie 0 or 5 stars, I said | 0, 5 |
| 13 | Input: Well rounded tribute.
Sentiment: Positive.

Input: Saw how bad this movie was.
Sentiment: | Positive, Negative |
| 14 | Review: Well rounded tribute.
Positive: True.

Review: Saw how bad this movie was.
Positive: | True, False |
| 15 | Review: Well rounded tribute.
Stars: 5.

Review: Saw how bad this movie was.
Stars: | 5, 0 |

