# OpenReview forum: "Overthinking the Truth: Understanding how Language Models process False Demonstrations"
_ICLR.cc/2023/Conference — Submitted to ICLR 2023_

### Official Review · Reviewer_PZwM · 2022-10-23

**Confidence:** 4
**Correctness:** 2
**Technical Novelty And Significance:** 3
**Empirical Novelty And Significance:** 3
**Recommendation:** 5

**Clarity, Quality, Novelty And Reproducibility:**

* The clarity, quality, novelty and reproducibility of the paper are all very good in general.
* “False demonstrations”: As I briefly mentioned earlier, I think the term “false demonstrations” might be too abstract and broad to describe what the paper actually does. It would be better to say “demonstrations with permuted labels”.


**Strength And Weaknesses:**

### Strength
* The topic is interesting and timely. It has a set of interesting analyses in in-context learning.
* It includes detailed analysis on how attention heads contribute to in-context learning by attending to labels in the demonstrations that share the same true label as the test input. In fact, as far as I know, this is the first work that shows how in-context learning works with respect to attention heads (while Olsson et al identified such induction heads, they focus on language modeling rather than in-context learning with real NLP datasets). This analysis is also supported by experiments that measure the impact of zero-ing those attention heads. It is also quite clever to use synthetic dataset (called “unnatural dataset” in the paper) to find false prefix-matching heads and use them for NLP datasets that are real tasks.


### Weaknesses
The entire paper is based on the assumption that the model has to be robust to false demonstrations, which is not convincing. First of all, the demonstrations that are called “false demonstrations” in the paper are hardly “false” in my opinion. They still preserve the valid mapping between inputs and (an abstract notion of) labels, but the surface form of the labels are permuted. For instance, false demonstrations in the binary sentiment classification (positive->negative, negative->positive) can be seen as true demonstrations whose defined task is “map a positive review to the word ‘negative’ and a negative review to the word ‘positive’”. Therefore, it is not undesirable that the model predicts a permuted label instead of a true label (e.g., predicts ‘negative’ to a positive review), since this is the valid task defined by the demonstrations. In fact, some analysis in the paper (e.g. Figure 2b) shows that the model performance against original true labels degrades only because the model accurately predicts the permuted true label. I would be more convinced by the idea if the false demonstrations were demonstrations with random labels, because then the task defined by the demonstrations is not the valid task.



**Summary Of The Paper:**

This paper focuses on in-context learning where the model performs a classification task without gradient updates by reading a few labeled examples as part of the input (demonstrations). Specifically, the paper consists of three parts: (1) false demonstrations, where each label in the demonstrations is re-assigned based on the permutation of labels (e.g. positive->negative, negative->positive), leads to significant performance drop, due to the model choosing the permuted true label instead of the true label – which is not desired, because we want our model to be robust to false demonstrations. (2) This is due to the presence of “false prefix-matching heads”, a subset of attention heads that assign high attention score to labels in the demonstrations paired with the input that share the true label as the test input, and they mostly present in the later layers of the Transformer. (3) Zero-ing out those attention heads significantly reduces the gap in performance between true demonstrations and false demonstrations.


**Summary Of The Review:**

In summary, I think this paper has a set of very interesting findings about in-context learning and novel analysis on the connection between attention heads and in-context learning. It is well-written and is very easy to follow. However, I have a critical concern in the most important underlying assumption in the paper – that the “false demonstrations” in the paper is not actually the false demonstrations but is rather a re-definition of the task, and it is unclear why the goal is to make the model predict the original, true label. If authors/other reviewers agree with this, this means the paper has to be almost completely rewritten: for instance, the authors can re-run the same experiments/analysis with random labels (instead of permuted labels) and likely the same findings will still hold; or alternatively, the paper can be re-written to be about how the model is able to perform the task when the task is redefined with permuted labels whose semantic meaning doesn’t match with pre-training, attribute this to a few attention heads, and prove it is true by showing zero-ing out such attention heads significantly degrades performance again redefined (permuted) true labels.

---

> ### Author Response · Authors · 2022-11-15
> **Response #4**
>
> Thank you for your helpful comments.
>
> If we understand correctly, your main concern with our submission is similar to Reviewer 1’s: given that the demonstration labels are simply permuted, it’s not clear that our setup shows the model’s failure on the original task, rather than its success on a modified task.
>
> We address this concern in three ways:
> * We run our experiments in different settings, such as (as you mentioned) when demonstration labels are random, and find consistent results
> * We explain why we believe that permuted labels are still a relevant failure mode
> * We argue that our results remain informative regardless of what one believes the model ‘should’ do
>
> First, we ran our experiments in two new setups: 1) choosing the demonstration labels at random and 2) choosing the demonstration labels at random among the incorrect labels (plot of logit lens accuracies for [AGNews](https://imgur.com/a/IB0v3vk), [DBPedia](https://imgur.com/a/xSrG6Kv), and [averaged over datasets](https://imgur.com/a/rZk5L72)): in these cases, the demonstration labels are no longer simply permuted. We will add these results to our edited draft.
>
> As you predicted, our results still hold in these settings:
> * There is still an accuracy gap between true and false prefixes: 20.9 percentage points in (1) and 40.1 percentage points in (2).
> * Early-exiting at layer 16 outperforms full model evaluation for 7/8 datasets in (1) and 8/8 datasets in (2).
> * The accuracy gap reaches 35% of its final value at layer 13-14 for 7/8 datasets in (1) and 7/8 datasets in (2).
> * On average, ablating 10 false-prefix matching heads reduces the accuracy gap by 48.7% in (1) and 40.7% in (2), with a small effect on the true prefix: -2 percentage points in (1) and -1.8 percentage points in (2) (plot of average logit lens accuracies with and without ablations in [our original setting](https://imgur.com/a/YoovfvO), for [(1)](https://imgur.com/a/o3gKpxU) and for [(2)](https://imgur.com/a/47D8VTU)).
>
> We also considered a third setup: using the correct labels for half of the demonstrations, and permuted labels for the other half.
> * Here too, there is still an accuracy gap between true and false prefixes: 35.6 percentage points.
> * Early-exiting at layer 16 still outperforms full model evaluation for all 8 datasets.
> * The accuracy gap reaches 35% of its final value at layer 13-14 for 6/8 datasets.
> * On average, ablating 10 false-prefix matching heads reduces the accuracy gap by 33.5%, with a small effect on the true prefix: -1.06 percentage points (plot of logit lens accuracies with and without ablations, for [SST-2](https://imgur.com/a/zMppXTm) and [averaged over datasets](https://imgur.com/a/8LPfr1F)).
>
> We agree that in our original setting, repeating the permutation of the labels is a predictable consequence of the language modeling objective, and that it could be construed as good performance at the task ‘follow the pattern in the context’.
> However, the labels are not arbitrary: there is still an underlying ground truth, which we think justifies calling the demonstrations ‘false’.
> Indeed, humans often err in consistent, systematic ways: this means that prompts with permuted labels resemble realistic failure modes.
> For example, suppose a naive coder is confused between 2 functions f1 and f2, so always inverts the 2 functions in the prompt.
> If a code completion model then outputs f1 where correct code would contain f2, this may be good performance at the task ‘follow the patterns in the user’s prompt’, but it’s still not good from the point of view of generating good code.
>
> Finally, our results are informative regardless of what one believes the model ‘should’ do given demonstrations with permuted labels.
> We chose this setting specifically because there is a conflict between two heuristics the model could follow: “following the pattern in the demonstrations” and “using the surface form of the labels, e.g. directly associating the word ‘great’ in a review to the label ‘Positive’”.
> Our early-exiting results show that even though the former heuristic ultimately dominates at the later layers, the latter heuristic leads to higher accuracies at the earlier layers.

---

> > ### Author Response · Authors · 2022-12-13
> > **Response #4 (Follow Up)**
> >
> > Thank you again for the helpful comments! We hope we addressed your concerns and look forward to hearing your feedback.

---

### Official Review · Reviewer_TBuw · 2022-10-24

**Confidence:** 3
**Correctness:** 3
**Technical Novelty And Significance:** 4
**Empirical Novelty And Significance:** 4
**Recommendation:** 6

**Clarity, Quality, Novelty And Reproducibility:**

Clarity: This paper is written clearly, reading it was enjoyable.
Quality: This paper is of high quality, the empirical evidence provided is very strong, the logical flow of the experiments and presentation is coherent and consistent.
Novelty: The problem itself is an obvious next step for AI Safety and alignment research, authors’ approach was mainly based on prior work, but the combination of prior methods with this problem is novel. This makes this paper somewhat novel but not significantly novel since there is little algorithmic innovation. But given the significance of the problem itself and the insights obtained, I still think this paper is above the ICLR standard.


**Strength And Weaknesses:**

Strength: Clarity, quality (see next section)
Weakness: technical novelty (see next section)


**Summary Of The Paper:**

This paper investigates what happens if LLMs are prompted with false demonstrations: the LLMs will output false answers. Authors further investigate the underlying mechanism at a neuron level for this phenomenon and discovered that early exiting would reduce wrong labels. Furthermore, authors identified induction heads that are responsible for outputting wrong labels, showing empirical evidence that LLMs are “post-processing” the truth at later layers, mainly with a few(~15) induction heads.

**Summary Of The Review:**

I recommend accept as I think this paper provided significant empirical insights on a significant problem, and the presentation of this paper is clear, and the logical flow is coherent.

---

> ### Author Response · Authors · 2022-11-15
> **Response #3**
>
> Thank you for your kind review.
>
> Let us know if you have any other questions about our submission.

---

> > ### Comment · Reviewer_TBuw · 2022-12-02
> > **Updates after discussions**
> >
> > I have read other reviews and responses and after discussing with other reviewers. Though the additional experiments on six new datasets have added more evidence that the analysis might generalize to new datasets, I agree that one major concern of this method is the generalizability of the analysis because even with the new single combined metric, deciding on which heads to ablate is still tasks dependent and not guaranteed to work well on any arbitrary tasks. This made me want to decrease my score.
> >
> > But upon more reflection, this might be too harsh a requirement, and furthermore there does seem to be some consistent patterns across datasets in terms of the accuracy gap versus exiting at which layer, supporting authors' claims. This made me want to maintain my score.
> >
> > All reviewers agreed that the task itself is at least somewhat interesting and revealed interesting insights, and the paper is well written.
> >
> > So in conclusion I decided to lower my score but not by too much, from 8 to 6, marginal accept.

---

> > > ### Author Response · Authors · 2022-12-06
> > > **Thanks for the Feedback**
> > >
> > > Thank you for the feedback! We appreciate the clear outline of your concerns, which we aim to highlight and address.
> > >
> > > > Though the additional experiments on six new datasets have added more evidence that the analysis might generalize to new datasets, I agree that one major concern of this method is the generalizability of the analysis because even with the new single combined metric, deciding on which heads to ablate is still tasks dependent and not guaranteed to work well on any arbitrary tasks. This made me want to decrease my score.
> > >
> > > **Data/task-dependent:**
> > >
> > > In our submission, we selected 10 heads on a toy dataset, unnatural. We ablate these exact 10 heads on 14 other datasets to demonstrate that any dataset (even a toy task) can be used to select heads, and that these heads generalize to other datasets.
> > >
> > > Ablating the 10 heads (selected on unnatural) were, in fact, effective on all 14 datasets spanning an array of tasks (i.e. sentiment analysis, NLI, topic classification, etc). The ablations were also robust to various choices of k (e.g. k = 10, k = 40).
> > >
> > > We also showed that our method does not depend on the number of heads to ablate. Concretely, in Figure 7, the gap reduction monotonically increases with the number of heads ablated.
> > >
> > > **Model Dependence**
> > >
> > > We agree that the ablation method is model dependent. However, many effective mitigation techniques are model dependent (e.g. adversarial training, fine-tuning, identifying attention heads responsible for gender bias, calibration, etc). We propose a general method, however, that can be run on any autoregressive LM. Our method has three properties that make it attractive:
> > >
> > > 1. Efficient: Only need a small number of forward passes (e.g. 100) on a single dataset to select the heads.
> > >
> > > 2. Robust: Any dataset can be used to select heads, since these heads will likely generalize to unseen tasks as we’ve shown.
> > >
> > > 3. 1-degree of freedom: There is only one hyperparameter: the number $k$ of heads to ablate. We highlighted in Figure 7 that various selections of k are effective (e.g. 5, 15, 35, etc). Once $k$ is chosen, the heads are then selected by our metric with no other consideration needed (in response to, “perform careful selection on the critical layers and heads”)
> > >
> > > **Setup Dependence**
> > >
> > > We demonstrated that all our main findings hold under different setups.
> > > 1. When 50% of the labels are permuted incorrect labels and the other 50% are correct labels.
> > > 2. When 100% of labels are randomly selected among the possible incorrect answers.
> > > 3. When 100% of labels are randomly selected among the set of all possible labels.
> > >
> > > We would like to note:
> > >
> > > The ablation method enjoys a performance gain given false demonstrations. However, we find it equally important in that it provides evidence of our claims:
> > > 1) There are a few false-prefix matching heads that are consistent across datasets.
> > > 2) These heads are responsible for driving the performance gap between true and false prefixes.
> > >
> > > Furthermore, we find the ablation method as one component in aiming to understanding properties of language models that would have been hard to predict ex ante:
> > > 1. Without requiring any additional training of a probe, we show how the earlier layers can compute the correct answer, before the later layers follow the patterns in the false context.
> > > 2. The accuracy gap between true and false demonstrations appears at the same layers across almost all datasets.
> > > 3. The accuracy gap between true and false demonstrations appear as the false-prefix matching heads also appear.

---

> > > > ### Author Response · Authors · 2022-12-13
> > > > **Request for Feedback**
> > > >
> > > > Thanks again for the helpful review! Please let us know if our response helps assuage your concerns. We look forward to hearing your feedback.

---

### Official Review · Reviewer_4Jp2 · 2022-10-24

**Confidence:** 4
**Correctness:** 3
**Technical Novelty And Significance:** 3
**Empirical Novelty And Significance:** 3
**Recommendation:** 5

**Clarity, Quality, Novelty And Reproducibility:**

Clarify: the paper is very clear.

Quality: the paper presents some interesting findings.

Novelty: the finding that later layers contributed more to wrong predictions under incorrect demonstrations is quite novel.

Reproducibility: no code is provided, not sure if the analysis in the paper can be reproduced.

**Strength And Weaknesses:**

Strengths:
- The findings that later layers contribute more to the errors under incorrect demonstrations are quite novel.
- The proposed method of zeroing-out later layers or certain heads seem to be effective in reducing the gaps between correct and incorrect demonstrations.

Weaknesses:
- I think the overall analysis presents quite interesting findings but the *robustness* and *generalizability* of the analysis is questionable. There are many arbitrary choices in performing the analysis/mitigation, e.g.,

1)  to ablate the heads in Section 4, for the head choice (Appendix A.1 has the details), the authors "first consider the 25 heads with the highest label-attending score, then select the 10 heads with the highest class-sensitivity score"; further some bias was added "towards heads in the later layers, like selecting the 5 heads of 25 that belong to layers 20 and later". How are those choices made? They look very arbitrary to me and I'm not sure if it can generalize to other models or other tasks.

2) the early-exit strategy in Figure 3, it's true that the overall trend shows later layers contribute more to the enlarged gaps, but the gap happens at different layers for different tasks. Zeroing out all layers after 16 might lead to an overall better performance, but for many tasks the performance under incorrect demos have already started degradation long before layer 16. In addition, this layer choice is based on the overall pattern analysis on these 8 datasets, would this conclusion generalize to a new task/dataset?

3) how would people apply the method in practice? From Figure 3, the accuracy under correct demos are highly sensitive to the #layers. Would early-existing sometimes cause significant performance drop if not applied carefully after the critical layer? In addition, it seems like people need to perform extensive analysis for each model/task combination to identify the critical layers / false prefix-matching heads (as they all seem different across models/tasks), how can one apply the method in a scalable way for any new/unseen tasks?

- In Table 1, any analysis to explain why the ablation doesn't work for natural language inference (SICK)? Are there more tasks in this category analyzed to show is it a single-task failure or the method doesn't work for the entire category?

- As an analysis paper, some of the key ablations are missing from the study, e.g., the false demos permute the label space for the entire demonstration. What if some of the labels remain correct and some of them are wrong? Would that change the conclusion significantly?

**Summary Of The Paper:**

This paper performs an extensive analysis on how language models suffer from incorrect demonstrations, and which layers/heads contribute to such performance degradations. The authors further propose a few ways for mitigating this problem, e.g., zeroing-out after critical layers, and ablating the false prefix-matching heads, and show they can effectively reduce the performance gap between correct and incorrect demos.

**Summary Of The Review:**

Overall I think this paper presents some quite interesting findings, in terms of under incorrect demonstrations, what's the contribution of each layer and how the gap between incorrect demos and correct demos can be minimized via zeroing-out certain heads. I do have some doubts in terms of the robustness of the analysis and its generalizability across different/unseen tasks, so not sure if the method can be applied generally to any task, which requires further study. Thus I recommend weak rejection for now.

---

> ### Author Response · Authors · 2022-11-15
> **Response #2**
>
> Thank you for your useful review.
>
> Our understanding is that your main concerns with the submission are about 1) our method for selecting which heads to ablate, 2) the consistency of critical layers across different datasets, 3) why ablations don’t reduce the accuracy gap for the SICK dataset, and 4) robustness to small changes in our experiment setup, such as some incorrect labels and some correct labels in the demonstrations.
>
> ### Selecting which heads to ablate
>
> > There are many arbitrary choices in performing the analysis/mitigation, e.g., to ablate the heads in Section 4, for the head choice (Appendix A.1 has the details), the authors "first consider the 25 heads with the highest label-attending score, then select the 10 heads with the highest class-sensitivity score";
>
> To address this concern, we repeated our experiments using a simpler metric to identify false prefix-matching heads: instead of thresholding first on the label-attending score, and then on the class-sensitivity score, we directly threshold on the following score, which directly incentivizes both label-attention and class-sensitivity:
>
> $$\\sum\\limits_i \\text{Att}_h(x, y_i) · 1[\\text{class}(x) = \\text{class}(x_i)] - \\frac {1} {\\# \\text{labels} - 1} \\sum\\limits_i \\text{Att}_h(x, y_i) · 1[\\text{class}(x) \\neq \\text{class}(x_i)]$$
>
> This way, we have a single metric instead of two, which removes a degree of freedom.
> We show the effect on SST-2 ([plot of logit lens accuracies](https://imgur.com/a/ZkxZGrc)) and on average over all datasets ([plot of logit lens accuracies](https://imgur.com/a/YoovfvO)) of ablating a very small fraction of all heads, the 10 heads with the largest scores on the toy ‘unnatural’ dataset. We find that ablating these heads results in a large reduction in the accuracy gap between true and false prefixes (28.5%), with a comparatively small effect on the performance given true prefixes (-1.4% change).
>
> > further some bias was added "towards heads in the later layers, like selecting the 5 heads of 25 that belong to layers 20 and later". How are those choices made? They look very arbitrary to me and I'm not sure if it can generalize to other models or other tasks.
>
> We biased ablating towards the later layers because they had the highest permutation score (see this [plot of the permutation score](https://imgur.com/a/fjeHUH8) averaged over datasets).
> This led to a greater reduction in the gap between true and false prefixes.
>
> However, as the above results show, the effect is still very clear when selecting heads using the metric alone, without any bias towards the later layers.
>
> Regarding generalization to other tasks, we ran the ablated model from the submission on 6 other datasets: Financial Phrasebank, ETHOS, TweetEval-Atheism, TweetEval-Feminist, Medical Question Pairs, MRPC, and Recognizing Textual Entailment.
> We found consistent results: [here](https://imgur.com/a/J4FCBq0) we show the extension of Table 1 to these new datasets.
> Averaging over the 14 datasets, we find that ablation reduces the accuracy gap by 23.2%, with a small effect on the performance given true prefixes (-2.2% change).

---

> > ### Author Response · Authors · 2022-11-15
> > **Response #2 (followed)**
> >
> > ### How consistent are the ‘critical layers’?
> >
> > > the early-exit strategy in Figure 3, it's true that the overall trend shows later layers contribute more to the enlarged gaps, but the gap happens at different layers for different tasks. Zeroing out all layers after 16 might lead to an overall better performance, but for many tasks the performance under incorrect demos have already started degradation long before layer 16. In addition, this layer choice is based on the overall pattern analysis on these 8 datasets, would this conclusion generalize to a new task/dataset?
> >
> > We disagree that ‘the gap happens at different layers for different tasks’.
> >
> > For GPT-J, the accuracy gap between true and false prefixes first exceeds:
> > * 45% of its final value at layers 13-14 for 12/14 datasets.
> > * 50% of its final value at layers 13-14 for 12/14 datasets.
> > * 55% of its final value at layers 13-14 for 12/14 datasets.
> >
> > This also applies to the other models.
> >
> > For GPT2-XL, the accuracy gap between true and false prefixes first exceeds:
> > * 45% of its final value at layers 21-23 for 11/14 datasets.
> > * 50% of its final value at layers 21-23 for 11/14 datasets.
> > * 55% of its final value at layers 21-23 for 11/14 datasets.
> >
> > For GPT-NeoX, the accuracy gap between true and false prefixes first exceeds:
> > * 45% of its final value at layers 10-12 for 13/14 datasets.
> > * 50% of its final value at layers 10-12 for 13/14 datasets.
> > * 55% of its final value at layers 10-12 for 13/14 datasets.
> >
> > We agree that the optimal decoding layer varies across datasets.
> >
> > Despite this, choosing layer 16 remains better than decoding at the final layer, and this generalizes to new tasks: when adding the 6 new datasets mentioned above, we find that exiting at layer 16 outperforms running the entire model for 12/14 datasets.
> >
> > > how would people apply the method in practice? From Figure 3, the accuracy under correct demos are highly sensitive to the #layers. Would early-existing sometimes cause significant performance drop if not applied carefully after the critical layer? In addition, it seems like people need to perform extensive analysis for each model/task combination to identify the critical layers / false prefix-matching heads (as they all seem different across models/tasks), how can one apply the method in a scalable way for any new/unseen tasks?
> >
> > We're thinking of this as a method to reveal knowledge that models represent but don't output, not as a way to improve SOTA. In particular, because it doesn’t require training any additional probes, there’s no need to perform extensive analysis to run the method: it’s sufficient to pick several layers and early-exit from each of them. We ran our extensive analysis to better understand the method, but it is not a requirement for early exiting.

---

> > > ### Author Response · Authors · 2022-11-15
> > > **Response #2 (followed)**
> > >
> > > ### Why do ablations not reduce the gap for SICK?
> > > > In Table 1, any analysis to explain why the ablation doesn't work for natural language inference (SICK)? Are there more tasks in this category analyzed to show is it a single-task failure or the method doesn't work for the entire category?
> > >
> > > Using the simpler head selection procedure we presented above, we find that ablating the 10 heads with the highest score on ‘unnatural’ reduces the gap on SICK: the gap is reduced by 29.5% for k = 40, and 38.4% for k=10. Therefore, the 15 heads we used in the original submission were suboptimal for SICK.
> > >
> > > We also test our ablations on another NLI task: RTE.
> > > The gap decreases by 81.8% for k=10, but increases by 25% for k=40.
> > > However, for k=39, the gap decreases by 62.5%, and similarly decreases for k=30-38.
> > > Unlike for SICK, the [gap](https://imgur.com/a/Wl2YkpA) for RTE is very small, which we believe explains these very noisy results.
> > >
> > > ### What if some labels remain correct, and some of them are wrong?
> > >
> > > > As an analysis paper, some of the key ablations are missing from the study, e.g., the false demos permute the label space for the entire demonstration. What if some of the labels remain correct and some of them are wrong? Would that change the conclusion significantly?
> > >
> > > We our experiments in two new setups: 1) choosing the demonstration labels at random and 2) choosing the demonstration labels at random among the incorrect labels (plot of logit lens accuracies for [AGNews](https://imgur.com/a/IB0v3vk), [DBPedia](https://imgur.com/a/xSrG6Kv), and [averaged over datasets](https://imgur.com/a/rZk5L72): in these cases, the demonstration labels are no longer simply permuted. We will add these results to our edited draft.
> > >
> > > As reviewer 4 predicted, our results still hold in these settings:
> > > * There is still an accuracy gap between true and false prefixes: 20.9 percentage points in (1) and 40.1 percentage points in (2).
> > > * Early-exiting at layer 16 outperforms full model evaluation for 7/8 datasets in (1) and 8/8 datasets in (2).
> > > * The accuracy gap reaches 35% of its final value at layer 13-14 for 7/8 datasets in (1) and 7/8 datasets in (2).
> > > * On average, ablating 10 false-prefix matching heads reduces the accuracy gap by 48.7% in (1) and 40.7% in (2), with a small effect on the true prefix: -2 percentage points in (1) and -1.8 percentage points in (2) (plot of average logit lens accuracies with and without ablations in our [original setting](https://imgur.com/a/YoovfvO), for [(1)](https://imgur.com/a/o3gKpxU) and for [(2)](https://imgur.com/a/47D8VTU)).
> > >
> > > We also considered another setup: using the correct labels for half of the demonstrations, and permuted labels for the other half.
> > > * Here too, there is still an accuracy gap between true and false prefixes: 35.6 percentage points.
> > > * Early-exiting at layer 16 still outperforms full model evaluation for all 8 datasets.
> > > * The accuracy gap reaches 35% of its final value at layer 13-14 for 6/8 datasets.
> > > * On average, ablating 10 false-prefix matching heads reduces the accuracy gap by 33.5%, with a small effect on the true prefix: -1.06 percentage points (plot of logit lens accuracies with and without ablations, for [SST-2](https://imgur.com/a/zMppXTm) and [averaged over datasets](https://imgur.com/a/8LPfr1F)).
> > >
> > > We will add these new results to the updated version of our submission.

---

> > > > ### Comment · Reviewer_4Jp2 · 2022-12-02
> > > > **Thanks for the response**
> > > >
> > > > Thanks the authors for the detailed response, and thanks for adding the proposal on the simpler metric, which is much easier to use compared to the selection criteria in the original paper.
> > > >
> > > > I still have some concerns regarding the mitigation strategy though, which is too much data/task-dependent and model-dependent, and if someone wants to apply it, they will need to redo the analysis from scratch and perform careful selection on the critical layers and heads. Also the results on SICK and RTE showed the performance can vary quite a bit depending on the choice of k, and the method is not guaranteed to work well on any arbitrary task.
> > > >
> > > > Overall I tend to keep my current rating, but I think the findings are quite interesting and the simpler metric could be potentially improved as a more general methodology for easier mitigations on an arbitrary task.

---

> > > > > ### Author Response · Authors · 2022-12-06
> > > > > **Follow Up Response**
> > > > >
> > > > > Thank you for the feedback! We appreciate the detailed comment, which we hope to address below.
> > > > >
> > > > > > I still have some concerns regarding the mitigation strategy though, which is too much data/task-dependent and model-dependent, and if someone wants to apply it, they will need to redo the analysis from scratch and perform careful selection on the critical layers and heads.
> > > > >
> > > > > **Data/task-dependent:**
> > > > >
> > > > > In our submission, we selected 10 heads on a toy dataset, unnatural. We ablate these exact 10 heads on 14 other datasets to demonstrate that any dataset (even a toy task) can be used to select heads, and that these heads generalize to other datasets.
> > > > >
> > > > > Ablating the 10 heads (selected on unnatural) were, in fact, effective on all 14 datasets spanning an array of tasks (i.e. sentiment analysis, NLI, topic classification, etc). The ablations were also robust to various choices of k (e.g. k = 10, k = 40).
> > > > >
> > > > > We also showed that our method does not depend on the number of heads to ablate. Concretely, in Figure 7, the gap reduction monotonically increases with the number of heads ablated.
> > > > >
> > > > > **Model Dependence**
> > > > >
> > > > > We agree that the ablation method is model dependent. However, many effective mitigation techniques are model dependent (e.g. adversarial training, fine-tuning, identifying attention heads responsible for gender bias, calibration, etc). We propose a general method, however, that can be run on any autoregressive LM. Our method has three properties that make it attractive:
> > > > >
> > > > > 1. Efficient: Only need a small number of forward passes (e.g. 100) on a single dataset to select the heads.
> > > > >
> > > > > 2. Robust: Any dataset can be used to select heads, since these heads will likely generalize to unseen tasks as we’ve shown.
> > > > >
> > > > > 3. 1-degree of freedom: There is only one hyperparameter: the number $k$ of heads to ablate. We highlighted in Figure 7 that various selections of k are effective (e.g. 5, 15, 35, etc). Once $k$ is chosen, the heads are then selected by our metric with no other consideration needed (in response to, “perform careful selection on the critical layers and heads”)
> > > > >
> > > > > **Setup Dependence**
> > > > >
> > > > > We demonstrated that all our main findings hold under different setups.
> > > > > 1. When 50% of the labels are permuted incorrect labels and the other 50% are correct labels.
> > > > > 2. When 100% of labels are randomly selected among the possible incorrect answers.
> > > > > 3. When 100% of labels are randomly selected among the set of all possible labels.
> > > > >
> > > > > We would like to note:
> > > > >
> > > > > The ablation method enjoys a performance gain given false demonstrations. However, we find it equally important in that it provides evidence of our claims:
> > > > > 1) There are a few false-prefix matching heads that are consistent across datasets.
> > > > > 2) These heads are responsible for driving the performance gap between true and false prefixes.
> > > > >
> > > > > Furthermore, we find the ablation method as one component in aiming to understanding properties of language models that would have been hard to predict ex ante:
> > > > > 1. Without requiring any additional training of a probe, we show how the earlier layers can compute the correct answer, before the later layers follow the patterns in the false context.
> > > > > 2. The accuracy gap between true and false demonstrations appears at the same layers across almost all datasets.
> > > > > 3. The accuracy gap between true and false demonstrations appear as the false-prefix matching heads also appear.

---

> > > > > > ### Author Response · Authors · 2022-12-13
> > > > > > **Request for Thoughts**
> > > > > >
> > > > > > Thanks again for your helpful review! Please let us know if our response helps assuage your concerns. We look forward to hearing your feedback.

---

> > > > > ### Author Response · Authors · 2022-12-06
> > > > > **Follow Up Response (continued)**
> > > > >
> > > > > We would also like to address these two points:
> > > > >
> > > > > > Also the results on SICK and RTE showed the performance can vary quite a bit depending on the choice of k, and the method is not guaranteed to work well on any arbitrary task.
> > > > >
> > > > > The performance varies quite a bit on RTE since it has a very small accuracy gap between true and false demonstrations (i.e. less than 1% accuracy gap for $k = 40$). Thus, the reduction of the gap when ablating will de facto yield somewhat difference results for varying $k$.
> > > > >
> > > > > With our updated metric for selecting heads, SICK yields qualitatively similar results for $k=10$ and $k=40$. The reduction in the accuracy gap for $k = 10$ is slightly large than $k=40$ since the accuracy gap is significantly smaller with 10 demonstrations -- and so any improvement will have a larger percentage reduction.
> > > > >
> > > > > Please let us know if you have any additional concerns.

---

### Official Review · Reviewer_gm7Z · 2022-10-26

**Confidence:** 4
**Correctness:** 3
**Technical Novelty And Significance:** 2
**Empirical Novelty And Significance:** 2
**Recommendation:** 5

**Clarity, Quality, Novelty And Reproducibility:**

The paper is clearly written. The methods mostly follow the existing "logit lens" and "induction heads" approaches, but apply them into their analysis. The findings from the analysis are generally useful but might not be novel enough.

**Strength And Weaknesses:**

Strengths:
* This paper takes a closer look at how the model will perform at different layers & attention heads when presented with incorrect labels. The findings generally confirm that PLMs are still largely impacted by incorrect labels, which is a bit contradictory to the messages in Min et al., 2022* but aligns with Yoo et al., 2022*. The experiments look convincing.
* It's also interesting that the method of "logit lens" can work with the hidden representation at different layers without further training. After zeroing out the later layers, the final output layer can still work.

Weaknesses:
* My biggest concern is on the underlying assumption of this paper---the model should not be impacted by the incorrect labels. It's controversial here because when we say in-context "learning", we actually want the model to be truthful to the provided demonstration examples. I am willing to hear the author's thoughts on this.
* Despite that this work might be the first to show that the copy of incorrect labels emerges in the later layers, this finding is not that surprising considering the early layers are mostly responsible for fusing information, and the later layers are mainly for generating the final output (see the references).
* Regarding the proposed method, although removing the particular components can mitigate the negative impact of incorrect labels, it's also questionable how this will affect the generation capability. It would be helpful if the author can show how much this will change the model's performance when the labels are correct.

References:

\* Sewon Min, Xinxi Lyu, Ari Holtzman, Mikel Artetxe, Mike Lewis, Hannaneh Hajishirzi, and Luke Zettlemoyer. Rethinking the role of demonstrations: What makes in-context learning work?

\* Kang Min Yoo, Junyeob Kim, Hyuhng Joon Kim, Hyunsoo Cho, Hwiyeol Jo, Sang-Woo Lee, Sang-goo Lee, Taeuk Kim. Ground-Truth Labels Matter: A Deeper Look into Input-Label Demonstrations

\* Nelson F. Liu, Matt Gardner, Yonatan Belinkov, Matthew E. Peters, Noah A. Smith. Linguistic Knowledge and Transferability of Contextual Representations

\* Jesse Vig, Yonatan Belinkov. Analyzing the Structure of Attention in a Transformer Language Model

**Summary Of The Paper:**

This paper studies how pretrained language models might be affected by **incorrect** in-context examples at different layers. By using the "logit lens" approach and analyzing the attention heads, they found such the copying or generation of these incorrect labels emerge in the later layers and some specific attention heads. After removing these components, they demonstrate the models are less affected by the incorrect labels.

**Summary Of The Review:**

This paper provides a deeper analysis of how PLMs perform at different layers and attention heads when presented with incorrect labels, and they find that generation of incorrect labels is mostly related to the later layers and specific heads. So, they propose to remove these components and show that this mitigates the effect of incorrect labels. Although it's interesting to see such an analysis of in-context learning, the findings are expected overall, and the proposed method cannot really compete with the performance of full language models.

---

> ### Author Response · Authors · 2022-11-15
> **Response #1**
>
> Thank you for your valuable feedback.
>
> If we understand correctly, your main concerns with the submission are about 1) whether it’s a failure of the model that it outputs the permuted labels, 2) whether the ablations affect the accuracy given correct demonstrations, and 3) how surprising our findings are.
>
> ### Should the model be impacted by the incorrect labels?
>
> > My biggest concern is on the underlying assumption of this paper---the model should not be impacted by the incorrect labels. It's controversial here because when we say in-context "learning", we actually want the model to be truthful to the provided demonstration examples. I am willing to hear the author's thoughts on this.
>
> A first concern that you mentioned is similar to Reviewer 4’s main concern: given that the demonstration labels are simply permuted, it’s not clear that our setup shows the model’s failure on the original task, rather than its success on a modified task.
>
> We address this in three ways:
> * We run our experiments in different settings, such as when demonstration labels are random, and find consistent results
> * We explain why we believe that permuted labels are still a relevant failure mode
> * We argue that our results remain informative regardless of what one believes the model ‘should’ do.
>
> We ran the same experiments in two new setups: 1) choosing the demonstration labels at random and 2) choosing the demonstration labels at random among the incorrect labels (plot of logit lens accuracies for [AGNews](https://imgur.com/a/IB0v3vk), [DBPedia](https://imgur.com/a/xSrG6Kv), and [averaged over datasets](https://imgur.com/a/rZk5L72)): in these cases, the demonstration labels are no longer simply permuted (except for 2-way tasks in (2)).
>
> As reviewer 4 predicted, our results still hold in these settings:
> * There is still an accuracy gap between true and false prefixes: 20.9 percentage points in (1) and 40.1 percentage points in (2).
> * Early-exiting at layer 16 outperforms full model evaluation for 7/8 datasets in (1) and 8/8 datasets in (2).
> * The accuracy gap reaches 35% of its final value at layer 13-14 for 7/8 datasets in (1) and 7/8 datasets in (2).
> * On average, ablating 10 false-prefix matching heads reduces the accuracy gap by 48.7% in (1) and 40.7% in (2), with a small effect on the true prefix: -2 percentage points in (1) and -1.8 percentage points in (2) (plot of average logit lens accuracies with and without ablations in [our original setting](https://imgur.com/a/YoovfvO), for [(1)](https://imgur.com/a/o3gKpxU) and for [(2)](https://imgur.com/a/47D8VTU)).
>
> We also considered another setup: using the correct labels for half of the demonstrations, and permuted labels for the other half.
> * Here too, there is still an accuracy gap between true and false prefixes: 35.6 percentage points.
> * Early-exiting at layer 16 still outperforms full model evaluation for all 8 datasets.
> * The accuracy gap reaches 35% of its final value at layer 13-14 for 6/8 datasets.
> * On average, ablating 10 false-prefix matching heads reduces the accuracy gap by 33.5%, with a small effect on the true prefix: -1.06 percentage points (plot of logit lens accuracies with and without ablations, for [SST-2](https://imgur.com/a/zMppXTm) and [averaged over datasets](https://imgur.com/a/8LPfr1F)).
>
> We will add these new results to the updated version of our submission.
>
> We agree that in our original setting, repeating the permutation of the labels is a predictable consequence of the language modeling objective, and that it could be construed as good performance at the task ‘follow the pattern in the context’.
> However, the labels are not arbitrary: there is still an underlying ground truth, which we think justifies calling the demonstrations ‘false’.
> Moreover, humans often err in consistent, systematic ways: this means that prompts with permuted labels resemble realistic failure modes.
> For example, suppose a naive coder is confused between 2 functions f1 and f2, so always inverts the 2 functions in the prompt.
> If a code completion model then outputs f1 where correct code would contain f2, this may be good performance at the task ‘follow the patterns in the user’s prompt’, but it’s still not good from the point of view of generating good code.
>
>
> Finally, our results are informative regardless of what one believes the model ‘should’ do given demonstrations with permuted labels.
> We chose this setting specifically because there is a conflict between two heuristics the model could follow: “following the pattern in the demonstrations” and “using the surface form of the labels, e.g. directly associating the word ‘great’ in a review to the label ‘Positive’”.
> Our early-exiting results show that even though the former heuristic ultimately dominates at the later layers, the latter heuristic leads to higher accuracies at the earlier layers.

---

> > ### Author Response · Authors · 2022-11-15
> > **Response #1 (followed)**
> >
> > ### Effect of ablations when the labels are correct
> >
> > > It would be helpful if the author can show how much this will change the model's performance when the labels are correct.
> >
> > We address this in the submission:
> > > While they greatly improve the accuracy given a false prefix, our ablations have a comparatively small effect on the accuracy given correct demonstrations: ablating the false prefix-matching heads decreases the accuracy given true demonstrations by 2% for k = 40 and by 0.31% for k = 10.
> >
> > Moreover, in Table 1, we showed the percentage difference in accuracy given true prefixes for all datasets, and Figure 6 shows the average logit lens accuracies of the original and lesioned models given true and false demonstrations.
> >
> > ### How expected are our findings?
> >
> > > Despite that this work might be the first to show that the copy of incorrect labels emerges in the later layers, this finding is not that surprising considering the early layers are mostly responsible for fusing information, and the later layers are mainly for generating the final output (see the references).
> >
> > For most datasets, including all the sentiment analysis and topic classification datasets, we find that the earlier layers are not just fusing information: they are already computing the correct answer.
> > For example, for GPT-J, on average over all datasets:
> > * At layer 2/28, the accuracy is already 4 percentage points above the random baseline.
> > * At layer 4/28, the accuracy is 11 percentage points above the random baseline.
> > * At layer 10/28, the accuracy is 17 percentage points above the random baseline.
> > * At layer 16/28, the accuracy is 24 percentage points above the random baseline.
> > * At the final layer 28/28, the accuracy is back to 4 percentage points above the random baseline.
> >
> > Moreover, Reviewer 2 believes that “The findings that later layers contribute more to the errors under incorrect demonstrations are quite novel.”
> >
> > > Although it's interesting to see such an analysis of in-context learning, the findings are expected overall
> >
> > We disagree that our “findings are expected overall”. For example, we believe the following findings are novel and informative, and would have been hard to predict ex ante:
> > * Without requiring any additional training of a probe, we show how the earlier layers can compute the correct answer, before the later layers follow the patterns in the false context.
> > * The accuracy gap between true and false demonstrations appears at the same layers across almost all datasets.
> > * Removing only a very small fraction of heads (10 out of 464) selected using a toy dataset significantly improves the accuracy given false demonstrations on a variety of datasets, with comparatively minimal effects on the accuracy given true demonstrations.

---

> > > ### Author Response · Authors · 2022-12-13
> > > **Thoughts on our Rebuttal?**
> > >
> > > Thanks again for your helpful review! We hope we addressed your concerns in our response, and we look forward to hearing your feedback.

---

### Decision · Program_Chairs · 2023-01-20

**Decision:**

Reject

**Justification For Why Not Higher Score:**

In the online meeting, all reviewers stated that they were happy with a reject decision.

**Justification For Why Not Lower Score:**

n/a

**Metareview: Summary, Strengths And Weaknesses:**

The paper explores the impact of providing false labels during few shot learning. It demonstrates that false labels can cause incorrect predictions from a language model, and shows that the effect can be mitigated by zero-ing certain heads in later layers that attend to the incorrect answers.

The paper is a borderline case. Reviewers disagreed somewhat with how relevant and interesting they find this experimental set up, how surprising the results are, different intuitions on what the model 'should' do on this task, and how well they expect the analysis to generalize to other datasets. The revision has partially addressed these concerns - however, the consensus in discussion was that the paper needs a bit more work before acceptance.


**Summary Of Ac-Reviewer Meeting:**

The paper represents a borderline case, and reviewers expressed legitimate differences in opinion over how interesting the experimental set up is, and how surprising the results are. There were also concerns about how well the method and analysis would generalize to other tasks. Overall, all reviewers stated that they were happy with a reject decision.